# POLCA: Stochastic Generative Optimization with LLM

**Xuanfei Ren** [1]   **Allen Nie** [2]   **Tengyang Xie** [*1]   **Ching-An Cheng** [*3]

## Abstract

Optimizing complex systems, ranging from LLM prompts to multi-turn agents, traditionally requires labor-intensive manual iteration. We formalize this as a stochastic generative optimization problem where a language model acts as an optimizer, guided by numerical and text feedback to discover the best system. We introduce **P**rioritized **O**ptimization with **L**ocal **C**ontextual **A**ggregation (POLCA), a scalable framework designed to handle stochasticity in optimization—such as noisy feedback, sampled minibatches, and stochastic system behaviors—while effectively managing the unconstrained expansion of solution space. POLCA maintains a priority queue to enable exploration-exploitation, tracking candidate solutions and their evaluation histories. To enhance efficiency, we integrate an $\varepsilon$-Net mechanism to maintain parameter diversity and an LLM Summarizer to perform meta-learning across historical trials. We theoretically prove that POLCA converges to near-optimal candidate solutions under stochasticity. We evaluate our framework on diverse benchmarks, including $\tau$-bench, HotpotQA, VeriBench, and KernelBench. Experimental results demonstrate that POLCA achieves robust, sample- and time-efficient performance, consistently outperforming state-of-the-art algorithms in both deterministic and stochastic problems. The codebase for this work is publicly available at https://github.com/rlx-lab/POLCA.

## 1. Introduction

Optimizing complex systems – from model prompts to code generators to agent harnesses – traditionally requires manual iteration by domain experts. Recently, generative optimiza-

tion has demonstrated successes in automating this process, showing successes in scientific discovery, code revision, and end-to-end system optimization (Cheng et al., 2024; Novikov et al., 2025; Agrawal et al., 2025). The solution to each of these problems can be viewed as a parameterized computer program.[1] Generative optimization is a process where a generative model, typically an LLM, modifies the parameter(s), validates the modification, and then revises it using feedback from validation. This process is repeated iteratively, either sequentially or under the coordination of a search algorithm, and the feedback can take the form of numerical scores, text, or multimodal signals. When provided with the right feedback (Pryzant et al., 2023; Chen et al., 2024; Xu et al., 2025), generative optimization algorithms can be significantly faster than black-box algorithms that use reward or preference as the only learning signals (Huang et al., 2023; Wei et al., 2025; Agrawal et al., 2025). Nonetheless, optimization instability and plateaus have also been reported, such as LLM repeating the same mistakes multiple times (Kumar et al., 2024; Chen et al., 2024).

The limitation arises when the LLM as the optimizer does not observe sufficient information from feedback to derive an improvement direction. This problem is exacerbated when obtaining a low-variance estimation of the parametrized program's performance is expensive, either because the environment is stochastic or because feedback from humans or ultra-large LLMs is costly. The former occurs when the program must be evaluated on many inputs or tasks to estimate its average performance, or when the program itself exhibits inherent stochastic behaviors. In such cases, obtaining high-quality feedback requires multiple evaluations, which can be too expensive to perform at every optimization step. Therefore, optimization often relies on stochastic estimates from only a few evaluations. On the other hand, language feedback from LLM judges or human users may be noisy and subjective as well. These challenges become even more severe when an LLM is used as the optimizer (Yang et al., 2023; Nie et al., 2024). Stochastic variation can cause the optimizer to generate many semantically similar parameters, so the search space grows linearly while semantically useful information does not (Shi et al., 2022;

---

*Corresponding authors. [1]University of Wisconsin-Madison [2]Google DeepMind [3]Google Research. Correspondence to: Tengyang Xie <tx@cs.wisc.edu>, Ching-An Cheng <chingan@google.com>.

*Proceedings of the 43rd International Conference on Machine Learning*, Seoul, South Korea. PMLR 306, 2026. Copyright 2026 by the author(s).

[1]The term *parameter* is used here to distinguish the changeable part of a program, as defined by the programmer. In the most general case, the entire program can be treated as a parameter.

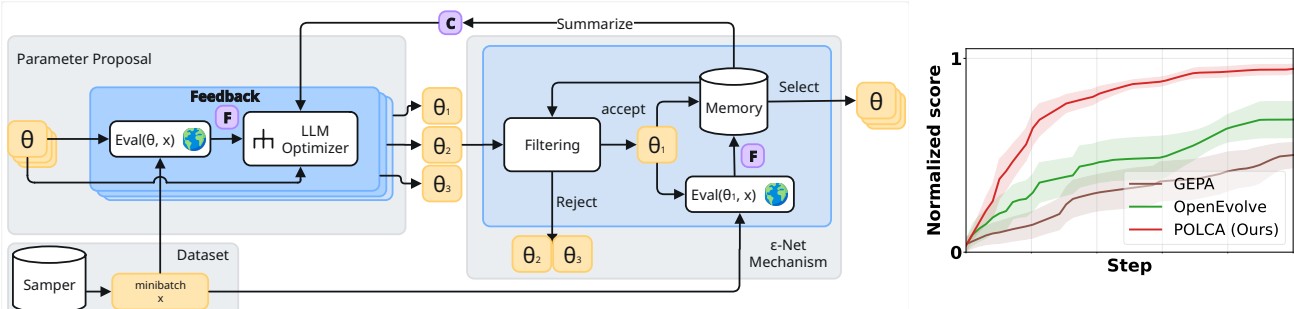

*Figure 1.* **Left**: The POLCA framework for generative optimization. POLCA maintains a memory buffer as an $\varepsilon$-Net to ensure diverse program storage. In each iteration, it selects promising parameter candidates from the $\varepsilon$-Net, evaluates them against a sampled minibatch, and generates new candidate parameters based on the feedback. These candidates undergo a semantic Filtering stage; accepted parameters are evaluated on the minibatch and integrated into the $\varepsilon$-Net. Finally, a Summarize step compresses the memory to provide concise global context $C$ for the next optimization cycle. **Right**: Normalized performance averaged across benchmarks ($\tau$-bench, HotpotQA, VeriBench, and KernelBench). The solid curve represents the mean, while the shaded region indicates the standard error across all benchmarks. Results are aggregated by standardizing scores and computational budgets to a scale of $[0, 1]$.

Li et al., 2022; Wang et al., 2025a; Lange et al., 2025a). Verifying these redundant programs requires additional evaluations, making generative optimization expensive to scale.

Take optimizing an LLM agent as an example, where the parameter may be its harness. The agent needs to handle a variety of task requests, and yet running the agent with each task takes minutes; therefore, only a subset of tasks can be used at a time for making an update in optimization. The feedback is likely generated by querying another LLM (using privileged task information or execution trace) (Yao et al., 2024; Lin et al., 2024; Wu et al., 2025), which can lead to stochastic evaluation and feedback. Lastly, the agent itself is stochastic as well because of the use of LLM inside. Other applications may manifest a subset of these stochasticity sources. If the outcome of the program is verifiable by a computer script, the scoring can be deterministic. If the inputs to the program can be efficiently enumerated (such as a small set of unit tests), sampling is not necessary. However, not all tasks are verifiable and LLM-as-a-Judge has become common (Zheng et al., 2023; Gu et al., 2024; Lee et al., 2025a). In addition, running the program on all inputs is not always practical as problems get complex, since this requires linear complexity per optimization step. It is desirable that algorithms can update based on sampled minibatches.

In this work, we introduce **P**rioritized **O**ptimization with **L**ocal **C**ontextual **A**ggregation (POLCA), a scalable framework for stochastic generative optimization (Figure 1). Under mild assumptions on LLM capabilities, we show that an embedding-based memory with an accept/reject rule, which we call the $\varepsilon$-Net criterion, can address two primary limitations of generative optimization: parameter update instability and evaluation stochasticity. This mechanism naturally bounds the total number of parameters that need to be evaluated. Without the $\varepsilon$-Net criterion, an LLM optimizer may propose new parameters indefinitely, resulting in

unbounded evaluation costs. Sufficient exploration of the parameter space is controlled by $\varepsilon$, which dictates a coverage-cost tradeoff, specified by the user. This is achieved by a reward-free embedding for each parameter that captures signals about the underlying reward. Such embeddings are increasingly available in modern LLMs (Lee et al., 2025b).

Incorporating search into generative optimization is common (e.g., evolutionary search (Novikov et al., 2025), Pareto frontier search (Agrawal et al., 2025), or beam search (Pryzant et al., 2023)). However, existing methods often assume an effectively unlimited evaluation budget or do not explicitly address evaluation stochasticity. In contrast, POLCA uses embeddings to construct the $\varepsilon$-Net criterion to decide whether a new parameter is sufficiently novel to explore. This mechanism yields a search process robust to stochasticity: it avoids discarding candidates too early while naturally bounding the number of distinct programs in the process. Prior work has also used filtering to promote novelty. For example, AlphaCode (Li et al., 2022) uses test-based filters, and ShinkaEvolve (Lange et al., 2025a) uses embedding-based filtering. However, these choices are empirically motivated. We show that an embedding-based mechanism is theoretically necessary for effective generative optimization under stochasticity.

Theoretically, we analyze POLCA with the UCB score as the priority for ranking candidate parameters. Under the assumption that the optimizer can achieve strict improvement within a certain reward range, we prove that POLCA eventually converges to near-optimal solutions under stochasticity. The convergence rate is primarily influenced by two factors: the efficiency of the optimization oracle, which captures the capability of the LLM optimizer, and the stochasticity of the evaluations, which is determined by the task. The former determines the number of iteration steps required to propose a near-optimal candidate, while the latter determines the

number of samples necessary to estimate the reward.

We conduct experiments to validate our algorithm across various sources of stochasticity. In $\tau$-bench (Yao et al., 2024), we optimize a tool-use LLM agent's prompts for multi-step problems. POLCA effectively handles stochasticity arising from both mini-batching and the agent's internal randomness, improving performance over multiple held-out tasks. It also achieves superior results in HotpotQA (Yang et al., 2018) prompt optimization. In formal verification (VeriBench) (Miranda et al., 2025), we test POLCA's ability to learn from compilation signals and stochastic LLM feedback when translating Python programs into Lean 4 code. We also validate POLCA in a deterministic setup using KernelBench (Ouyang et al., 2025) to optimize CUDA kernel code. Across all benchmarks, POLCA consistently outperforms state-of-the-art baselines, GEPA (Agrawal et al., 2025) and OpenEvolve (Sharma, 2025) – an open-source implementation of AlphaEvolve (Novikov et al., 2025), in both convergence speed and final performance. We additionally verify that parametric reward modeling via an ensemble is prone to evaluation stochasticity. This highlights the need to use a persistent memory mechanism such as $\varepsilon$-Net to robustly scale up LLM optimizers in the face of stochasticity.

## 2. Problem Setup

The optimization problem of a complex system using generative models can be formulated as an abstract problem, which we call *stochastic generative optimization* of a parameterized program $P_\theta$. Our goal is to design a scalable algorithm to automate this pipeline and to address the instability caused by stochasticity in sampling inputs, evaluations, and the program itself. While our primary focus in the experiments will be on prompt and code optimization, as we will show below, this formulation is generic and extends to broader domains including general discrete structures and functional solutions. Traditionally these tasks relied on a manual expert loop where human researchers iteratively refined parameters. By treating these systems as programs with parameters, we take a unified approach to handle stochasticity inherent in trial-and-error cycles.

**Problem Formulation**    We denote a stochastic generative optimization problem as a tuple $\{P, \theta_0, \mathcal{D}, \Theta, \mathcal{G}, \mathcal{O}\}$. The parameterized program $P_\theta$ is a mapping that takes input $x$ and returns output $y \sim P_\theta(x)$. We aim to change parts of the program $\theta$ which we call the *parameter*. The parameter can include numerical values, text strings, codes, or a mixture of them. We denote $\theta_0$ as the initial parameter, and let $\Theta$ represent all possible program parameters. Our goal is to improve the program's performance on a data distribution $\mathcal{D}$. Each data point $(x, \omega) \sim \mathcal{D}$ contains an input $x$ to the program and some associated side information $\omega$. Given $(\omega, x, y)$, there is an oracle $\mathcal{G}$, which we call the

*guide*, to provide a numerical score $r \in \mathbb{R}$ and feedback $f$ (e.g., error messages, critiques, or gradients) (with stochasticity) to guide the optimization process. In other words, the score $r$ and feedback $f$ implicitly encapsulate the optimization objective, just as gradients in first-order optimization. Lastly, we suppose there is an LLM Optimizer $\mathcal{O}$ that can propose new parameters after seeing $(\theta, x, y, r, f, c)$, where $c$ denotes additional context about the optimization problem. This LLM Optimizer acts as an oracle to propose new candidate parameters $\theta' \in \Theta$ by interpreting the guide's feedback, but we do not assume $\theta'$ would always be better than $\theta$. Failure to improve may be due to the lack of information in $(\theta, x, y, r, f, c)$ or due to the stochasticity of the LLM Optimizer.

**Evaluation and Objective**    The performance of a program $P_\theta$ is characterized by its expected reward $\mu(\theta)$, defined as: $\mu(\theta) = \mathbb{E}_{\omega, x \sim \mathcal{D}} \left[ \mathbb{E}_{y \sim P_\theta(x)} [\mathcal{G}_r(\omega, y, x)] \right]$, where $\mathcal{G}_r$ denotes the score $r$ returned by the guide. Given a computational resource budget (e.g. number of evaluation metric calls), we wish to design an algorithm **Alg** to maximize the expected reward of the selected agent: $\max_{\textbf{Alg}} \mathbb{E}[\mu(\theta_{\text{best}})]$, where $\theta_{\text{best}}$ denotes the best parameter **Alg** returns, and the expectation is taken over the joint stochasticity of the data sampling of $\mathcal{D}$, the program $P$ and the LLM optimizer $\mathcal{O}$, the evaluations provided by the guide $\mathcal{G}$, and the inherent randomness of **Alg**. The algorithm coordinates the interactions between the dataset $\mathcal{D}$, the guide $\mathcal{G}$, and the LLM optimizer $\mathcal{O}$ to balance between exploring novel parameters and accurately identifying high-performing candidates amidst stochastic variations due to them.

**Expansive Search Space**    One important property is that the parameter space $\Theta$ is often exponentially large and discrete without a natural ordering (e.g., the space of all valid Python programs). This property makes this optimization setting differ significantly from standard scenarios such as finite-arm bandits, where a learner either has access to the full action set up front. In contrast, the action space (namely the parameter space) here can only be accessed through querying the LLM optimizer $\mathcal{O}$, and the proposal distribution of the LLM optimizer is highly dependent on the specific information in $(\theta, x, y, r, f, c)$ provided at each iteration. Consequently, to effectively optimize, we also need to select meaningful improving parameter $\theta$ and data $x$, curate good context $c$ (e.g., by integrating historical evaluations and feedback) in order to steer the LLM optimizer toward generating increasingly superior candidates.

## 3. Algorithm

In this paper, we design a generative optimization algorithm, **P**rioritized **O**ptimization with **L**ocal **C**ontextual **A**ggregation (POLCA), which utilizes a continuously updated memory and an $\varepsilon$-Net criterion to handle stochasticity

from program evaluations. We emphasize that POLCA is a *search algorithm built around a given optimizer*: the LLM-driven optimizer $\mathcal{O}$ in Section 2 is treated as an external proposal mechanism beyond our control, and POLCA orchestrates how to allocate evaluations, manage memory, and route context to that oracle. This separation is intentional—it makes POLCA interoperable with future, stronger LLM optimizers without requiring any modification to the search procedure itself, and is what allows the analysis in Section 4 to characterize the search algorithm cleanly without having to model the internal behavior of the LLM.

The optimization procedure is formally presented in Algorithm 1. We maintain a priority queue $\mathcal{Q}$—initialized with the base program $\theta_0$—which functions as a memory and is continuously updated with empirical results. Each iteration of the optimization loop begins by sampling a minibatch $\mathcal{B} \subset \mathcal{D}$ and selecting a subset of the empirically best-performing programs, $\Theta_{\text{explore}} \subset \mathcal{Q}$. We then collect data $\mathcal{S}$ by evaluating each $\theta \in \Theta_{\text{explore}}$ on $\mathcal{B}$; these results are used to directly update the performance statistics in $\mathcal{Q}$. Subsequently, the optimizer $\mathcal{O}$ utilizes the newly collected data $\mathcal{S}$, in conjunction with a broader context $c_{\text{history}}$, to propose a set of raw program parameters $\Theta_{\text{raw}}$. Here, $c_{\text{history}}$ is provided by another external LLM component called the *Summarizer*, which processes the entire updated priority queue $\mathcal{Q}$ to generate high-level optimization instructions for the optimizer $\mathcal{O}$. To prevent the memory $\mathcal{Q}$ from being overwhelmed by semantically similar candidates, $\Theta_{\text{raw}}$ is filtered through an $\varepsilon$-Net-based semantic filter operation to obtain $\Theta_{\text{new}}$. This filtering mechanism constrains the size of the parameter space while ensuring structural and semantic diversity. Finally, the candidates in $\Theta_{\text{new}}$ are evaluated on the same minibatch $\mathcal{B}$ to obtain initial performance estimates before being added to the memory. In the following sections, we elaborate on the specific mechanics of each component and their contributions to the optimization process.

**Minibatch Evaluation**    The scale of the dataset $\mathcal{D}$ presents a primary bottleneck, when doing a full evaluation of $P_\theta$ in every iteration is computationally out of reach. This makes obtaining a precise, reliable score prohibitively expensive for every generated candidate. Instead, minibatch sampling is employed to estimate the performance of proposed programs and provide valuable feedback for further optimization. We implement a SAMPLEMINIBATCH process in POLCA, where in each iteration, a minibatch of tasks $\mathcal{B} = \{(\omega_i, x_i)\}_{i=1}^{B}$ is randomly sampled from the dataset $\mathcal{D}$ with replacement. A program $P_\theta$ evaluated on $\mathcal{B}$ yields stochastic observations $\{(\theta, \omega_i, x_i, y_i, r_i, f_i)\}_{i=1}^{B}$. The stochasticity in the scores of $P_\theta$ arises from the minibatch sampling, the program execution, and the guide evaluation, as discussed in Section 2. The same minibatch evaluation is performed for both $\Theta_{\text{explore}}$ and the newly proposed $\Theta_{\text{new}}$ to ensure a fair comparison, thereby mitigating

---

**Algorithm 1** POLCA

**Require:** Dataset $\mathcal{D}$, base agent $\theta_0$, Guide $\mathcal{G}$, Optimizer $\mathcal{O}$
**Ensure:** Best program $\theta_{\text{best}}$ identified during search
1:  **Initialize:** $\mathcal{Q} \leftarrow \{\theta_0\}$
2:  **while** Budget not exhausted **do**
3:      $\mathcal{B} \leftarrow \text{SAMPLEMINIBATCH}(\mathcal{D})$
4:      $\Theta_{\text{explore}} \leftarrow \text{SELECTPROGRAMS}(\mathcal{Q})$
5:      $\mathcal{S} \leftarrow \text{EVALUATE}(\Theta_{\text{explore}}, \mathcal{B}, \mathcal{G})$
6:      $\mathcal{Q} \leftarrow \text{UPDATESTATS}(\mathcal{Q}, \mathcal{S})$
7:      $\Theta_{\text{raw}} \leftarrow \text{PROPOSEPROGRAMS}(\mathcal{O}, \mathcal{S}, \mathcal{Q})$
8:      $\Theta_{\text{new}} \leftarrow \text{SEMANTICFILTER}(\Theta_{\text{raw}}, \mathcal{Q})$
9:      $\mathcal{S} \leftarrow \text{EVALUATE}(\Theta_{\text{new}}, \mathcal{B}, \mathcal{G})$
10:     $\mathcal{Q} \leftarrow \text{UPDATESTATS}(\mathcal{Q}, \mathcal{S})$
11: **end while**
12: **return** $\theta_{\text{best}} \in \mathcal{Q}$ with the highest empirical mean score

---

potential bias arising from task-specific variance. Both evaluation processes are fully parallelized[2], which we elaborate on further in Algorithm 2 in Section C.

**Priority Queue Memory**    To handle the stochasticity in program evaluation, we maintain $\mathcal{Q}$ as a *priority queue*. For each program $\theta \in \mathcal{Q}$, we assign an exploration *priority* derived from its data. Specifically, for a program $\theta$ with data $\{(\theta, w_n, x_n, y_n, r_n, f_n)\}_{n=1}^{N}$, we define the priority as the empirical mean score[3]: $p_{\text{explore}}(\theta) = (\sum_{n=1}^{N} r_n)/N$. To identify programs for improvement at the start of each iteration of POLCA, we invoke $\Theta \leftarrow \text{SELECTPROGRAMS}(\mathcal{Q})$ to retrieve the candidates with the highest $p_{\text{explore}}$. During the optimization process, newly collected data $\mathcal{S} = \{(\theta, w, x, y, r, f)\}$ is integrated via $\text{UPDATESTATS}(\mathcal{Q}, \mathcal{S})$. This function updates the priorities for all relevant $\theta \in \mathcal{Q}$ and reorders the queue to facilitate efficient exploration. This architecture directly addresses the three sources of stochasticity by continuously updating the dynamic priority queue $\mathcal{Q}$. Under this design, programs with superior performance are evaluated consecutively, and $p_{\text{explore}}(\theta)$ eventually converges to the true expected reward $\mu(\theta)$ as variance is averaged out. This design ensures that promising candidates with temporarily low means can be revisited and refined later, while those with low potential are eventually deprioritized after sufficient sampling.

**Generative Parameter Space Growth**    We assume our generative optimizer oracle $\mathcal{O}$ has a proposal distribution $\Pi(\cdot \mid \mathcal{C})$, where $\mathcal{C}$ represents the input context. To enhance the optimization trajectory, we first utilize an external LLM called the *Summarizer* to aggregate the history of

---

[2]When program execution or guide evaluation rely heavily on LLM calls, this parallelization becomes significantly more efficient, as the parallelization of LLM API calls can be easily implemented.

[3]Other priority functions can be plugged in for alternative strategies (Section D). Replacing the mean with UCB score yields a provable guarantee (Section 4).

successes and failures from $\mathcal{Q}$ into a $c_{\text{history}}$, providing high-level context for the optimizer. Then, for each parameter $\theta \in \Theta_{\text{explore}}$, the optimizer is invoked using the minibatch collected in this iteration, $\mathcal{S}_\theta = \{(\theta, \omega_i, x_i, y_i, r_i, f_i)\}_{i=1}^B$, augmented by $c_{\text{history}}$ to synthesize information from previous iterations. In this formulation, utilizing only the current minibatch $\mathcal{S}_\theta$ is analogous to a standard *first-order update* in numerical optimization. By incorporating $c_{\text{history}}$ from the Summarizer, the process mirrors *Momentum-based methods* (Cui et al., 2024), leveraging the past evaluations to stabilize the search and escape local optima. In parallel, the optimizer analyzes the local context $\mathcal{S}_\theta$ and the global summary $c_{\text{history}}$ to propose a candidate $\theta'$ designed to achieve superior performance: $\theta' \sim \Pi(\cdot \mid \mathcal{C}_\theta)$, where $\mathcal{C}_\theta = \{(\theta, x_i, y_i, r_i, f_i, c_{\text{history}})\}_{i=1}^B$. We collect the proposed program parameters as $\Theta_{\text{raw}}$. A more detailed description of the program generation process can be found in Algorithm 3 and Section C.

**Semantic Filtering based on $\varepsilon$-Net**   While we tackle the stochasticity of individual program evaluations by using a continuously updated memory $\mathcal{Q}$, indiscriminately adding all new programs to the memory would cause $\mathcal{Q}$ to grow linearly with the number of iterations. This growth can lead to prohibitive sample complexity when attempting to identify the best program. This can be avoided, because the input context to $\mathcal{O}$ often exhibits comparatively low variance, due to *1)* minibatches may overlap or repeat, and *2)* specific programs (or highly similar ones) are repeatedly selected for exploration. Consequently, LLM-based optimizers tend to propose many semantically similar parameters over time, meaning the growth of useful information in $\mathcal{Q}$ does not scale at the same rate as the number of programs.

To navigate this complexity, we leverage the latent semantic structure of the parameter space to discretize $\Theta$. Let $\phi : \Theta \to \mathbb{R}^d$ be an embedding function that maps parameters into a dense vector space. We then define a semantic distance metric $\tilde{d}(\theta, \theta') = \|\phi(\theta) - \phi(\theta')\|_2$ to measure the semantic similarity between any two parameters $\theta, \theta' \in \Theta$. All newly generated program parameters $\theta' \in \Theta_{\text{raw}}$ are subsequently processed by the SEMANTICFILTER component. This component ensures that the priority queue $\mathcal{Q}$ is maintained as an $\varepsilon$-Net, such that any two programs in memory maintain a distance greater than $\varepsilon$. A new program parameter is admitted only if its semantic distance to every existing parameter in $\mathcal{Q}$ exceeds $\varepsilon$, thereby pruning redundant proposals and maintaining population diversity. The diversity within $\mathcal{Q}$ also facilitates the retrieval of a high-quality historical context $c_{\text{history}}$, as a more semantically diverse $\mathcal{Q}$ provides a more representative set of observations to the Summarizer. Detailed implementation of this filtering process is provided by Algorithm 4 in Section C.

## 4. Theoretical Analysis

In this section, we theoretically analyze POLCA (Algorithm 1). For clarity, we analyze a simplified version that selects one program and proposes one new program per iteration. Unlike Section 3, this version uses the UCB score rather than the empirical mean as the priority, since optimistic exploration is needed to certify provable guarantees. The mean priority used in practice is not in tension with this analysis: our ablation in Section E.6 shows that adding a UCB bonus brings only marginal empirical gains, justifying the simpler mean priority as the default.

Assume the reward function $\mu : \Theta \to [0, B]$, and that for any $\theta \in \Theta$ the score observation $r(\theta)$ is $\sigma^2$ sub-Gaussian with mean $\mu(\theta)$. The simplified POLCA starts from $\theta_0$ and runs for a horizon of $n$ steps. At step $t$, let $T_\theta(t)$ be the number of observations of $\theta$ and $\widehat{\mu}_{\theta,s}$ be its empirical mean over the first $s$ observations. The algorithm selects the program with the highest UCB score $UCB_\theta(t) = \widehat{\mu}_{\theta, T_\theta(t)} + 2\sigma\sqrt{\log(t)/T_\theta(t)}$ to obtain a new observation $r(\theta)$, and the optimizer proposes a new candidate $\theta'$ from the local context $\mathcal{C}_\theta$; if $\theta'$ passes the $\varepsilon$-Net filter, it is evaluated once to get $r(\theta')$. By design of the $\varepsilon$-Net, the number of distinct programs evaluated is bounded by $N_\varepsilon$ (depending only on $\Theta$ and $\varepsilon$), which we use throughout the analysis.

We introduce Assumption 1, which assumes the optimizer makes a $\gamma$-strict improvement with positive probability.

**Assumption 1** (Strict improvement). *There exist constants $\gamma > 0$ and $\delta_0 \in (0, 1)$. For any $\theta \in \Theta$ with $\mu(\theta) \leq B - \gamma$, we assume the optimization oracle satisfies:* $\mathbb{P}_{\theta' \sim \Pi(\cdot|\mathcal{C}_\theta)}[\mu(\theta') > \mu(\theta) + \gamma] \geq \delta_0$.

Based on Assumption 1, in this generative optimization problem, we only have control over improving programs with rewards in $[0, B - \gamma]$, whereas we lack a guarantee from the given optimizer for proposing better programs when the seed program has a reward in the range $(B - \gamma, B]$. We view this assumption as the minimal abstraction needed to analyze any search algorithm built around an external optimizer: without some form of local-progress condition, the oracle could be adversarial—always returning worse candidates—and no search procedure on top of it can succeed. Rather than modeling the LLM optimizer's internal behavior, we treat $(\gamma, \delta_0)$ as the two parameters that summarize its quality, and the bound below makes explicit how they affect the sample complexity of POLCA. Stronger optimizers correspond to larger $\gamma$ and $\delta_0$ and tighter bounds.

Let $\Theta_t \subset \Theta$ denote the set of programs accepted during the first $t$ iterations. Theorem 1 demonstrates that POLCA with a UCB priority function converges to near-optimal programs with rewards in $[B - \gamma, B]$, which the optimizer cannot be guaranteed to improve further.

**Theorem 1.** *Suppose $\mu : \Theta \to [0, B]$. If we run POLCA*

*with the UCB priority for $n$ iterations, then the expected total number of selections for programs with rewards in $[0, B - \gamma]$ is bounded by*

$$\mathbb{E}\left[\sum_{\theta \in \Theta_n : \mu(\theta) \leq B - \gamma} T_\theta(n)\right] \lesssim \left(\frac{B}{2\gamma\delta_0} + \frac{64\sigma^2 N_\varepsilon}{\gamma^2}\right) \log(n).$$

When the reward observations are deterministic ($\sigma = 0$), the bound becomes independent of the program space and depends only on the reward space and the optimization oracle: $\mathbb{E}\left[\sum_{\theta \in \Theta_n : \mu(\theta) \leq B - \gamma} T_\theta(n)\right] \lesssim B \log(n)/(2\gamma\delta_0)$.

We provide the complete proof of Theorem 1 in Section B, but we can intuitively interpret the two terms in the upper bound. The first term, $B \log(n)/2\gamma\delta_0$, represents the number of iterations required to generate a near-optimal program with reward in $[B - \gamma, B]$ under the uncertainty of the optimizer, as described in Assumption 1. The second term, with order $O(\sigma^2 N_\varepsilon \log(n)/\gamma^2)$, is the number of samples needed to estimate the expected reward of each program given stochastic observations. In the deterministic case, only one sample is needed to determine the reward of each program $\theta$, so the second term vanishes.

Theoretical analysis highlights the advantages of POLCA in maintaining a comprehensive historical record. Specifically, our $\varepsilon$-Net-based semantic filter prevents the memory buffer from growing linearly by rejecting semantically redundant candidates, ensuring the search remains scalable.

A naive implementation of generative optimization typically relies on sequential updates, where each iteration evaluates a single program to propose its successor. Below, we compare it with POLCA under the assumption of deterministic reward observations, where the algorithm has access to $\mu(\theta)$ for each generated $\theta$. Formally, at each step $t$, a *sequential updating* algorithm generates a new proposal based on the most recent observation: $\theta'_t \sim \Pi(\cdot \mid \mathcal{C}_{\theta_{t-1}})$. In contrast, POLCA updates by improving upon the best program found thus far: $\theta'_t \sim \Pi(\cdot \mid \mathcal{C}_{\tilde{\theta}_t})$, where $\tilde{\theta}_t := \arg\max_{\theta \in \Theta_t} \mu(\theta)$. Theorem 2 compares the rates at which updating rules yield a near-optimal program.

**Theorem 2.** *Suppose $\mu : \Theta \to [0, B]$ and the reward observation is deterministic. Under Assumption 1, the expected number of steps for a sequential updating algorithm* (4) *to propose a program with reward in $(B - \gamma, B]$ is $\mathcal{O}(1/\delta_0^{B/\gamma})$. In contrast, POLCA with updating rule* (5) *requires $B/(\gamma\delta_0)$ expected steps to reach the same threshold.*

The proof of Theorem 2 is provided in Section B.4. By using the historical maximum, POLCA maintains a monotonically non-decreasing reward baseline. This ensures the algorithm is robust to the stochasticity of the optimizer, as poor proposals cannot reset its progress.

## 5. Experiments

We implement POLCA within the Trace workflow optimization pipeline (Cheng et al., 2024), utilizing OptoPrime as the optimizer to conduct generative optimization guided by rich feedback and execution traces. We compare POLCA against established baselines (DSPy (Khattab et al., 2023), GEPA (Agrawal et al., 2025), and OpenEvolve (Sharma, 2025)); see Section E.1 for a detailed discussion on baselines. We select representative domains where stochasticity arises from minibatch sampling, program execution (Section 5.1), and evaluation methods (Section 5.2). Finally, we demonstrate the superiority of POLCA in deterministic domains in Section 5.3.

**Comparison criterion** Evaluating proposed programs in the search process is computationally expensive and time-consuming. While the total number of metric calls represents the actual computation used, we define an *evaluation step* as a unit where all constituent metric calls are parallelized, effectively measuring the number of sequential operations required as a surrogate of wall clock time. To ensure a fair comparison, we set a maximum budget of metric calls for all algorithms and report the scores achieved at each step. In Section E.3, we discuss criteria to evaluate and compare algorithms across multiple dimensions of wall-clock time and computational cost.

### 5.1. Stochasticity from program execution and minibatch sampling

One popular application of generative optimization involves training LLM-based agents. Since LLM-based agents are inherently stochastic, multiple trials on the same task may yield diverse outcomes. Furthermore, given the large number of potential tasks, evaluating agents on the entire training set is often inefficient. A widely used alternative is to sample a minibatch from the dataset to estimate agent performance. Consequently, the observed scores during the training process are inherently stochastic.

**$\tau$-bench** We first demonstrate such stochasticity using $\tau$-bench (Yao et al., 2024), a benchmark designed to evaluate agents in interacting with human users and executing tools to solve complex queries. Here we use `gemini-2.0-flash`[4] as the backbone model, `gemini/text-embedding-004`[5] as the embedding model for POLCA. The environment provides a sparse, binary reward $r \in \{0, 1\}$ per execution, indicating whether the user's request was resolved. We utilize the first 10 tasks from the retail domain of $\tau$-bench for optimization, with the remaining 105 tasks *held out* to test for generalization. The base agent provided

---

[4]https://docs.cloud.google.com/vertex-ai/generative-ai/docs/models/gemini/2-0-flash

[5]https://ai.google.dev/gemini-api/docs/embeddings

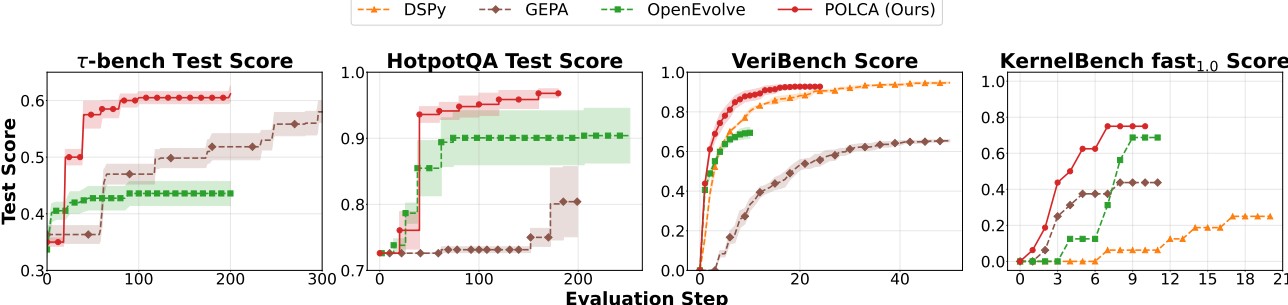

*Figure 2.* Search efficiency across four benchmarks ($\tau$-bench, HotpotQA, VeriBench, KernelBench). Solid curves represent the average highest score attained at each step, while the shaded regions denote the standard error across multiple independent runs (6 seeds for $\tau$-bench, 3 for HotpotQA and VeriBench, and 1 for KernelBench). Higher curves indicate superior efficiency.

*Table 1.* Pass@1 of $\tau$-**bench** retail domain (115 tasks). Prompts are trained on only the first 10 tasks. Here our POLCA achieves a 13% improvement compared with the base prompt.

| Method | First 10 tasks | Last 105 tasks | All 115 tasks |
|---|---|---|---|
| Base Prompt | 0.348 | 0.392 | 0.389 |
| GEPA | 0.557 | 0.417 | 0.429 |
| OpenEvolve | 0.373 | 0.422 | 0.418 |
| **POLCA (Ours)** | **0.575** | **0.425** | **0.439** |

by the benchmark is parameterized via a string variable, `additional_instructions`, which is appended to the system prompt. Details can be found in Section E.2.

Figure 2 demonstrates the effectiveness of POLCA. While OpenEvolve and GEPA demonstrate improvements over the base agent, they are significantly outperformed by POLCA in this stochastic environment. This discrepancy arises because these methods evaluate each proposed program on 10 tasks once without continuously updating the statistics, making them heavily sensitive to the stochastic evaluation.

We also evaluate the best generated prompt on the complete $\tau$-bench retail domain dataset, consisting of 115 tasks. Table 1 shows that the prompt generated by POLCA is not only the most effective on the 10-task training set but also achieves the best performance on the entire dataset.

**HotpotQA** HotpotQA (Yang et al., 2018) is a multi-hop question answering dataset where each task requires reasoning across multiple context paragraphs to produce a short answer. Each task consists of a question, 10 context paragraphs (of which 2–3 are relevant and the rest are distractors), and a ground-truth answer. We use `gemini-2.5-flash-lite` as the backbone model, and `gemini-embedding-001` as the embedding model for POLCA. Correctness is determined by exact match or substring containment, yielding a binary reward of 0 or 1 per task. We use different algorithms to optimize the prompt for question answering. Details can be found in Section E.2. The result in Figure 2 shows the superiority of POLCA.

### 5.2. Stochasticity from the evaluator

In the previous subsection, we discussed the stochasticity arising from program execution and the minibatch sampling tasks. Here we study the case of a stochastic evaluator for optimizing deterministic programs on a single task.

**VeriBench (3-step evaluation)** We consider the *formal verification* domain using VeriBench (Miranda et al., 2025), which evaluates the capability of LLMs to translate Python programs into verifiable Lean 4 code, with `claude-3.5-sonnet`[6] as the backbone and `gemini/text-embedding-004` as the embedding model. LLMs are prompted to translate the Python program into a compilable and semantically correct Lean 4 program. We formalize this problem by treating the entire translated Lean 4 program as the parameter, i.e., $P_\theta(x) \equiv \theta$, resulting in a fully deterministic program execution. The provided 3-step evaluation process of VeriBench, which contains compilation, unit tests, and an LLM judge, has stochasticity and provides a $[0, 1]$ reward for each proposed Lean 4 program. See Section E.2 for details.

Figure 2 shows that our algorithm outperforms all baselines, suggesting superior performance within the same time budget. In such cases, the evaluator is the only source of stochasticity. POLCA addresses this by continuously updating empirical mean scores, ensuring scalability compared to approaches using static performance values.

Algorithms such as DSPy and OpenEvolve typically collect feedback for a program only once, even if that data is used multiple times to generate new programs. In contrast, POLCA repeatedly selects high-performing programs to collect data; this not only helps for accurate estimation but also gathers diverse, stochastic feedback for these programs. Due to this stochasticity, obtaining feedback multiple times on the same parameter can increase the probability of receiving useful information to propose better programs. While

---

[6]https://www.anthropic.com/news/claude-3-5-sonnet

GEPA also collects data multiple times for promising programs, it remains limited because it: *(1)* depends highly on the initial validation and does not update program scores when new data is collected; *(2)* cannot explore in parallel; and *(3)* degenerates into always selecting the single best performer for exploration in single-task optimization problems, as the Pareto frontier collapses.

## 5.3. Deterministic Domains

Many generative optimization problems are nearly deterministic. Examples include code generation with a deterministic verifier and various scientific discovery problems. This class of problems is of equal significance in the field of generative optimization. We show that POLCA can be directly applied to fully deterministic domains without modification.

**VeriBench (Compilation)**
We utilize VeriBench again and the same LLMs but focusing on the deterministic compilation stage only. This remains challenging given the limited Lean 4 programming knowledge inherent in current LLMs. For this analysis, the reward is a binary indicator of compilation success; all other experimental settings remain unchanged. We provide comprehensive experimental details in Section E.2.

*Table 2.* **VeriBench** compilation pass rates. Each algorithm is allocated a budget of 50 metric calls per task.

| Algorithm | Pass Rate |
|---|---|
| DSPy | $0.888 \pm .023$ |
| GEPA | $0.695 \pm .010$ |
| OpenEvolve | $0.738 \pm .010$ |
| **POLCA (Ours)** | **$0.952 \pm .005$** |

The results for VeriBench compilation are presented in Table 2, with extended results available in Section E.4. The results show that DSPy, OpenEvolve, and GEPA are effective but remain less efficient than our methods. Our POLCA consistently outperforms these baselines, even when compared with the fully sequential DSPy and GEPA algorithms. Our algorithm reaches a 95.2% compilation pass rate (133/140) using a budget of 50 metric calls per task, significantly exceeding the baseline results. Our work represents the first thorough study applying agentic search algorithms to VeriBench; previous methods relied on simple iterative refinement and achieved considerably lower success rates. Specifically, Miranda et al. (2025) employed a sequential search with the same model (5 retries) on a subset of our tasks, achieving a maximum pass rate of 0.593 (67/113).

**KernelBench**  CUDA kernel optimization is also a popular problem suitable for generative optimization. We pick 16 matrix multiplication tasks from KernelBench (level 1) (Ouyang et al., 2025), which appear simple but remain challenging. As mentioned in Yan et al. (2026), these tasks are already highly optimized, making it difficult to achieve further speedups. We utilize the $\text{fast}_p$ score (Ouyang et al., 2025) defined as: $\text{fast}_p = \frac{1}{N} \sum_{i=1}^{N} \mathbb{1}(\text{correct}_i \wedge \{\text{speedup}_i > p\})$, where $p$ is the speedup threshold relative to the PyTorch

baseline. This metric measures the proportion of tasks for which the algorithm proposes a correct kernel with a speedup exceeding $p$. We utilize the `claude-3.7-sonnet`[7] model for generation and `gemini-embedding-001` as the embedding model. Figure 2 illustrates $\text{pass}_{1.0}$ performance comparisons[8] where POLCA distinctly outperforms the baselines. Details are provided in Section E.2.

The success of POLCA in deterministic domains can be attributed to two factors. First, the use of parallel starting points exploits the stochasticity of the optimizer more effectively than sequential baselines. Second, the context $c_{\text{history}}$ is summarized from programs of different paths. In contrast, DSPy, GEPA, and OpenEvolve are limited to local knowledge, focusing on a small set of optimization paths.

**Generality across model families.**  Across the four benchmarks above, our experiments cover `gemini-2.0-flash`, `gemini-2.5-flash-lite`, `claude-3.5-sonnet`, and `claude-3.7-sonnet`. To further demonstrate that POLCA is not tied to a particular model family, we additionally evaluate it with the open-source backbones `gpt-oss-20b` on HotpotQA and `Qwen3.5-122B-A10B` on VeriBench (Compilation). Even with these weaker open-source models, POLCA consistently outperforms the baselines; full results are deferred to Figures 14 and 15.

## 5.4. Ablation study

**Ablation on $\varepsilon$-Net and Summarizer**  In POLCA, we employ an $\varepsilon$-Net to filter programs and a Summarizer to provide a global context summary. We conduct an ablation study on these components; see Figure 3(a) (further comparisons across different metrics can be found in Figure 12), where vanilla POLCA refers to the version without the $\varepsilon$-Net or Summarizer. Comparing these variants highlights the advantages of our proposed components. Both the $\varepsilon$-Net and the Summarizer significantly improve performance over vanilla POLCA. In this domain, the $\varepsilon$-Net leverages embedding information to filter new candidates that are semantically similar to programs already in memory, thereby conserving a substantial portion of the sampling budget. Consequently, it achieves higher scores with fewer samples. The Summarizer enhances the optimizer by providing a broader context, utilizing the entire memory rather than just local observations to identify success and failure patterns across diverse programs, leading to the discovery of superior candidates.

**Ablation on $\varepsilon$ sensitivity**  We perform an ablation on the $\varepsilon$ value to provide intuition on how it affects performance on $\tau$-bench and VeriBench. The results are presented in Figures 3(b) and 3(c) (more ablation over different metrics can be found in Figure 13). By construction, the $\varepsilon$ value

---

[7]https://www.anthropic.com/news/claude-3-7-sonnet
[8]We provide a $\text{pass}_{0.5}$ analysis in Section E.5.

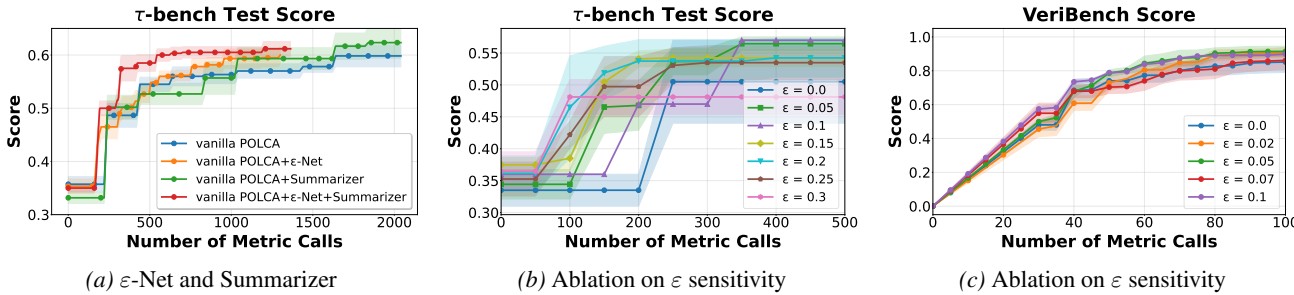

*(a) ε-Net and Summarizer*      *(b) Ablation on ε sensitivity*      *(c) Ablation on ε sensitivity*

*Figure 3.* Ablation study. Solid curves show the mean highest score achieved at each step, with shaded areas representing the standard error over independent runs (6 seeds for (a); 3 seeds for (b, c)).

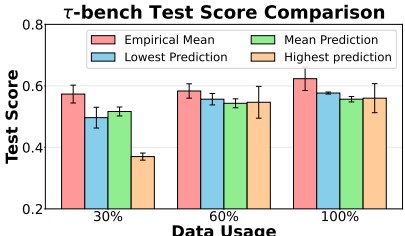

*Figure 4.* **Why not use regression?** Bar heights denote test scores for programs selected via different criteria across varying training data percentages. Results are averaged over 3 runs, with error bars indicating the standard error.

controls the coarseness of the discretization: parameters with a distance less than $\varepsilon$ are identified as the same by the algorithm. The ablation results show that POLCA's performance is not very sensitive to the exact $\varepsilon$ value within a certain range. We consistently find that $\varepsilon = 0$ (no discretization) yields the worst learning performance. For large $\varepsilon$, we observe a degradation in asymptotic performance due to approximation error of coarser discretization, although it improves the initial learning speed as expected. As $\varepsilon$ increases, POLCA accepts more diverse programs, thereby encouraging exploration across distinct program structures. When a reasonable $\varepsilon$ value is selected, it improves speed while incurring only negligible approximation error.

**Why not use regression?** In large-scale experiments, a substantial number of programs can be generated. If they are treated as independent, evaluating them all (multiple times) to identify optimal candidates would require an impractical sampling budget. We use an $\varepsilon$-Net filter with empirical mean to address this issue. We evaluate the feasibility of an alternate approach of training a surrogate reward function that maps program embeddings to predicted scores. We train an ensemble of five logistic regressors with semantic embedding vectors to predict scores of candidates generated by an optimization process in $\tau$-bench. We then select the best candidates based on the ensemble's *highest*, *mean*, and *lowest* predicted scores. As shown in Figure 4, these function approximators failed to outperform the simple selection

based on the *empirical mean* used in POLCA. A likely reason for this failure is that predicting a program's score accurately is difficult without explicit problem-instance information, even though embeddings do carry meaningful local information: a t-SNE visualization in Figure 16 shows that programs with similar embeddings tend to share similar scores, which is what the $\varepsilon$-Net filter exploits. The local smoothness, however, does not extend to globally accurate score prediction; see Section E.8 for details.

## 6. Conclusion

We formalize the problem of *stochastic generative optimization*, addressing the challenges of stochasticity in optimization and the unconstrained growth of the program space. We design POLCA, which introduces two primary contributions to address these challenges: (1) a continuously updated memory buffer that employs mean-based priorities to average out evaluation variance, and (2) a semantic $\varepsilon$-Net filtering mechanism that prunes redundant candidates to maintain a diverse and efficient search space. Theoretical analysis shows POLCA with such design could converge to near-optimal candidates efficiently. Empirical evaluations on $\tau$-bench, HotpotQA, VeriBench and KernelBench, covering both stochastic and deterministic domains, demonstrate that our approach significantly outperforms existing baselines, effectively handling stochasticity from minibatch sampling, program execution and evaluation.

**Limitations** Despite these advancements, POLCA has limitations. First, while the empirical mean is an efficient priority, more sophisticated strategies may exist. Although our analysis of POLCA with the UCB score has a good guarantee, it relies on the knowledge of the degree of stochasticity in the reward, and the assumption on the optimizer may not always be realistic. In addition, more advanced function approximation methods than embedding-based filtering are possible, e.g., by predicting performance across analogous tasks. Lastly, the observations made in the experimental results may be limited to the benchmarks and models tested here, despite our best efforts to make them representative.

## Acknowledgements

We acknowledge support of the DARPA AIQ Award and the Gemini Academic Program Award.

## Impact Statement

This paper presents work whose goal is to advance the field of Machine Learning. There are many potential societal consequences of our work, none which we feel must be specifically highlighted here.

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

# A. Related Work

Existing algorithms are primarily distinguished by their *optimizer design*, *search strategy*, and *candidate curation*. The optimizer design pertains to the specific method of invoking an LLM to generate a program parameter; the search strategy refers to the orchestration of the comprehensive optimization system, which may involve diverse and multiple LLM calls; whereas *candidate curation* addresses the principled filtering and selection mechanisms required to scale the process when the pool of generated programs exceeds computational or context limits.

**Optimizer Design**   Regarding optimizer design, some works utilize few-shot prompting, while others focus on sequential revision based on local observations such as rewards, textual feedback, and execution traces (Khattab et al., 2023; Cheng et al., 2024; Yuksekgonul et al., 2024). A more robust approach involves leveraging global knowledge by summarizing history to enhance performance (Cui et al., 2024). In large-scale search, context summarization is becoming increasingly popular (Zhang et al., 2025b). For example, in kernel optimization tasks, Lange et al. (2025b); Liao et al. (2025); Zhang et al. (2025a) employ an external LLM as a context summarizer to facilitate learning. POLCA similarly integrates numerical and textual feedback and leverages historical context summarization.

**Search Strategies**   Our work focuses on search strategies that balance exploration and exploitation. We maintain a program memory, where the exploration-exploitation trade-off involves both proposing improved programs via the optimizer and reducing uncertainty regarding our currently accessible programs. While basic methods rely on repeated generation and $N$-best selection, more sophisticated frameworks utilize in-context learning from rewards, iterative refinement, or task-merging (Khattab et al., 2023; Cheng et al., 2024). Specialized approaches include Beam Search (Sun et al., 2023; Pryzant et al., 2023; Chen et al., 2024), Monte-Carlo Tree Search (Wang et al., 2023), and Gibbs Sampling (Xu et al., 2024). Among these, Chen et al. (2024) proposes learning a reward model from collected data; while effective in certain domains, this may fail in the presence of highly stochastic reward observations. Some prompt optimization works (Pryzant et al., 2023; Cui et al., 2024) propose a finite-arm *bandit selection* phase, which is effective for small-scale search. However, this remains an exploitation-heavy method that may prematurely cease generating new programs. Conversely, difficult problems require continuous exploration. GEPA (Agrawal et al., 2025), designed for prompt tuning, maintains a Pareto frontier of undominated programs to preserve diversity. However, it is susceptible to stochastic observations, as it falsely rejects candidates. Furthermore, for single-task optimization with a verifier, GEPA may not be suitable, as Pareto-frontier-based search tends to degenerate, where the frontier merely represents the current best program. In such domains, AlphaEvolve (Novikov et al., 2025) utilizes MAP-Elites and island-based models to guide evolution. Subsequent frameworks like ThetaEvolve (Wang et al., 2025b) integrate evolutionary search with test-time reinforcement learning, while ShinkaEvolve (Lange et al., 2025a) employs rejection sampling and bandit-based ensemble selection. While these evolution-based methods excel at generating complex programs, they primarily address tasks with nearly deterministic verifiers and lack specific mechanisms to handle environments where the evaluation process is stochastic. Unlike prior methods that perform simple validation to reject non-improving candidates (Khattab et al., 2023; Novikov et al., 2025; Agrawal et al., 2025), POLCA manages stochasticity by continuously updating its memory buffer, including the re-evaluation of older candidates, as new information emerges. This allows the algorithm to perpetually learn and revisit candidates that demonstrate potential, mitigating the risk of false rejection due to noise.

**Candidate Curation**   One of the primary challenges in generative optimization is the tendency of LLMs to propose semantically redundant candidates during long-horizon search processes. Often, *clustering and filtering mechanisms* are used to maintain memory diversity and novelty. For example, AlphaCode (Li et al., 2022) utilizes test-based methods to reject underperforming candidates and clusters programs based on execution behavior to assess novelty and reduce evaluation need. Recent frameworks such as Kim et al. (2025) and Wang et al. (2025a) leverage embedding-based clustering to identify and collapse redundant reasoning states, thereby significantly pruning the search space while maintaining high optimization accuracy. Similarly, ShinkaEvolve (Lange et al., 2025a) employs embedding-based similarity detection coupled with an LLM-based code-novelty judge to accept or reject candidates. POLCA proposes the *semantic $\varepsilon$-Net filtering mechanism*, where under mild assumptions on the quality of the embedding (i.e., it contains useful information about the reward), we can control $\varepsilon$ to trade off between final proposed candidate's performance and search budget in a principled manner, while all previous works rely on multiple heuristic hyper-parameters without clear implications.

# B. Full Proof of Section 4

## B.1. Notations and the Technical Lemma

We run POLCA for $n$ steps. Without loss of generality, we assume that $B$ is divisible by $\gamma/2$. To begin our analysis, we partition the reward range into small intervals of width $\gamma/2$: $[0, B] = \bigcup_{k=1}^{2B/\gamma}[(k-1)\gamma/2, k\gamma/2] = [0, \gamma/2] \cup [\gamma/2, \gamma] \cup \cdots \cup [B - \gamma, B]$.

Define $I_k = \{\theta \in \Theta : \mu(\theta) \in [(k-1)\gamma/2, k\gamma/2]\}$ for any $1 \le k \le 2B/\gamma - 2$. We define $\tau_k$ to be the stopping time when the total number of selections in $I_k$ reaches $u_{\text{interval}} := 2\log(n)/\delta_0$, i.e.,

$$\tau_k := \min\left\{t : \sum_{\tilde{\theta} \in I_k} T_{\tilde{\theta}}(t) = u_{\text{interval}}\right\}. \tag{1}$$

Later we will show that this level of interval-level exploration is sufficient to propose a better program. For a single program $\theta \in I_k$, we focus on the additional selections occurring after this time ($t \ge \tau_k$), defined as:

$$T_\theta^{\text{add}}(t) := T_\theta(t) - T_\theta(\tau_k),$$

which represents the number of selections of an individual program after the interval itself has been sampled $u_{\text{interval}}$ times.

Define $u_{\text{single}} := \frac{64\sigma^2}{\gamma^2}\log(n)$. Our Theorem 3 shows that this additional number of selections can be bounded.

**Lemma 3** (Bounded selection number for each interval). *Let $\Theta_n$ denote the set of programs proposed and accepted in the first $n$ steps. Consider the case where $\theta \in \Theta_n$ and $\tau_k \le n$, such that $T_\theta^{add}(n)$ is well-defined. For any $\theta \in I_k$, the expected number of additional selections is bounded by:*

$$\mathbb{E}[T_\theta^{add}(n) \mid \theta \in \Theta_n, \tau_k \le n] \le u_{single} + 3 = \frac{64\sigma^2}{\gamma^2}\log(n) + 3.$$

We provide the proof of Theorem 3 in Section B.2. Theorem 3 serves as the most critical component of the entire proof. Intuitively, it demonstrates that POLCA with a UCB priority function will successfully distinguish between two programs with a reward gap of $\gamma$ after $O\left(\frac{\sigma^2}{\gamma^2}\log(n)\right)$ observations.

## B.2. Proof of Theorem 3

This lemma analyzes the additional selection number for any fixed $\theta \in I_k$. Define $\tau_\theta$ to be the time step at which the additional selections of program $\theta$ reach $u_{\text{single}}$, i.e., $\tau_\theta = \min_t\{T_\theta^{\text{add}}(t) = u_{\text{single}}\}$. Let $\tilde{\theta}$ be the first proposed program satisfying $\mu(\tilde{\theta}) > (k+1)\gamma/2$.

Define the "good event" $G = G_1 \cap G_2 \cap G_3$ as:

- **Concentration of program $\theta$:** $G_1 = \left\{\hat{\mu}_{\theta, T_\theta(\tau_\theta)} + 2\sigma\sqrt{\frac{\log(n)}{T_\theta(\tau_\theta)}} < (k+1)\gamma/2\right\}$

- **Proposal of $\tilde{\theta}$:** $G_2 = \{\text{the strictly better program } \tilde{\theta} \text{ is proposed before iteration } \tau_k\}$

- **Concentration for program $\tilde{\theta}$:** $G_3 = \{UCB_{\tilde{\theta}}(t) \ge \mu(\tilde{\theta}) \text{ for all } t \in [n]\}$

We first claim that when $G$ occurs, $T_\theta^{\text{add}}(n) \le u_{\text{single}}$ holds. Otherwise, if $T_\theta^{\text{add}}(n) > u_{\text{single}}$, there must exist $\tau_\theta < n$ such that $T_\theta^{\text{add}}(\tau_\theta) = u_{\text{single}}$. At iteration $\tau_\theta$, $T_\theta^{\text{add}} > 0$ implies $\tau_k \le \tau_\theta$. By $G_2$, a program $\tilde{\theta}$ with $\mu(\tilde{\theta}) > (k+1)\gamma/2$ has already been proposed. By $G_1$ and $G_3$, it follows that for all $\tau_\theta \le t \le n$:

$$UCB_{\tilde{\theta}}(t) \ge \mu(\tilde{\theta}) > (k+1)\gamma/2 > \hat{\mu}_{\theta, T_\theta(\tau_\theta)} + 2\sigma\sqrt{\frac{\log(n)}{T_\theta(\tau_\theta)}} = UCB_\theta(t).$$

Since the algorithm selects the program with the highest UCB score at each iteration, the fact that $UCB_{\tilde{\theta}}(t) > UCB_\theta(t)$ for all $t \ge \tau_\theta$ implies that $\theta$ will no longer be selected after time step $\tau_\theta$. This yields a contradiction to the assumption $T_\theta^{\text{add}}(n) > u_{\text{single}}$.

**Bounding** $P(G_1^c)$    Consider any fixed $s \geq u_{\text{single}}$. Since $u_{\text{single}} = \frac{64\sigma^2}{\gamma^2} \log(n)$, we have $2\sigma\sqrt{\frac{\log(n)}{s}} \leq \gamma/4$. We first bound $\mathbb{P}(G_1^c \mid T_\theta(\tau_\theta) = s)$:

$$
\begin{aligned}
\mathbb{P}(G_1^c \mid T_\theta(\tau_\theta) = s) &= \mathbb{P}\left( \hat{\mu}_{\theta, T_\theta(\tau_\theta)} + 2\sigma\sqrt{\frac{\log(n)}{T_\theta(\tau_\theta)}} \geq (k+1)\gamma/2 \mid T_\theta(\tau_\theta) = s \right) && \text{(Definition of } G_1) \\
&= \mathbb{P}\left( \hat{\mu}_{\theta,s} + 2\sigma\sqrt{\frac{\log(n)}{s}} \geq (k+1)\gamma/2 \right) \\
&= \mathbb{P}\left( \hat{\mu}_{\theta,s} - \mu(\theta) \geq (k+1)\gamma/2 - 2\sigma\sqrt{\frac{\log(n)}{s}} - \mu(\theta) \right) \\
&\leq \mathbb{P}\left( \hat{\mu}_{\theta,s} - \mu(\theta) \geq \frac{\gamma}{4} \right) && (\mu(\theta) \leq k\gamma/2) \\
&\leq \exp\left( -\frac{s\gamma^2}{32\sigma^2} \right) && \text{(Hoeffding's inequality)} \\
&\leq \exp\left( -\frac{u_{\text{single}}\gamma^2}{32\sigma^2} \right) && (s \geq u_{\text{single}}) \\
&\leq \frac{1}{n^2}.
\end{aligned}
$$

This probability bound has no relationship with the specific value of $T_\theta(\tau_\theta)$, so generally we have

$$
\mathbb{P}(G_1^c) \leq \frac{1}{n^2}.
$$

**Bounding** $\mathbb{P}(G_2^c)$    By the definition of $\tau_k$ in (1), at iteration $\tau_k$ the interval $I_k$ has been selected $u_{\text{interval}}$ times. The event $G_2^c$ implies that no $\gamma$-strictly better program $\tilde{\theta}$ has been generated after $u_{\text{interval}}$ selections on $I_k$. By Assumption 1, when we select a program $\theta \in I_k$ with reward $\mu(\theta) \in [(k-1)\gamma/2, k\gamma/2]$, the optimization oracle produces a $\theta'$ with $\mu(\theta') > (k+1)\gamma/2$ with probability at least $\delta_0$. Therefore, $\mathbb{P}(G_2^c)$ is the probability of failing to propose such a program for $u_{\text{interval}}$ consecutive trials:

$$
\mathbb{P}(G_2^c) \leq (1-\delta_0)^{u_{\text{interval}}} \leq e^{-\delta_0 u_{\text{interval}}} \leq e^{-\delta_0 \frac{1}{\delta_0} \log(n^2)} \leq \frac{1}{n^2}.
$$

**Bounding** $P(G_3^c)$    $G_3$ describes $n$ concentration bounds hold simultaneously.

$$
G_3^c = \bigcup_{t=1}^{n} \{UCB_{\tilde{\theta}}(t) < \mu(\tilde{\theta})\}.
$$

For a fixed $\theta_1 \in \Theta$,

$$
\begin{aligned}
\mathbb{P}(G_3^c \mid \tilde{\theta} = \theta_1) &= \mathbb{P}\bigg( \bigcup_{t=1}^{n} \{UCB_{\tilde{\theta}}(t) < \mu(\tilde{\theta})\} \mid \tilde{\theta} = \theta_1 \bigg) \\
&= \mathbb{P}\bigg( \bigcup_{t=1}^{n} \{UCB_{\theta_1}(t) < \mu(\theta_1)\} \bigg) \\
&\leq \sum_{t=1}^{n} \mathbb{P}(UCB_{\theta_1}(t) < \mu(\theta_1)) \\
&= \sum_{t=1}^{n} \mathbb{P}\bigg( \widehat{\mu}_{\theta, T_{\theta_1}(t)} + 2\sigma \sqrt{\frac{\log(n)}{T_{\theta_1}(t)}} < \mu(\theta_1) \bigg) \\
&= \sum_{t=1}^{n} \mathbb{P}\bigg( \widehat{\mu}_{\theta, T_{\theta_1}(t)} - \mu(\theta_1) < -2\sigma \sqrt{\frac{\log(n)}{T_{\theta_1}(t)}} \bigg) \\
&\leq \sum_{t=1}^{n} \frac{1}{n^2} = \frac{1}{n}. \qquad \text{(Hoeffding's inequality)}
\end{aligned}
$$

Since the final bound has no relationship with the value of $\theta_1$, we have

$$
\mathbb{P}(G_3^c) \leq \frac{1}{n}.
$$

Combining the bound for $\mathbb{P}(G_1^c), \mathbb{P}(G_2^c), \mathbb{P}(G_3^c)$, we have

$$
\mathbb{P}(G^c) \leq \frac{1}{n^2} + \frac{1}{n^2} + \frac{1}{n} = \frac{n+2}{n^2}.
$$

Strictly speaking, to obtain our final result, we should bound the probability given the event $\{\theta \in \Theta_n, \tau_k \leq n\}$. However, we can calculate the probability $\mathbb{P}(G_i^c \mid \Theta = \tilde{\Theta}, \tau_k = s)$ for fixed $\tilde{\Theta}, s$ and $i \in \{1, 2, 3\}$ using identical steps. Because the resulting bounds are independent of the value of $s$, our bound also holds in the conditional version:

$$
\mathbb{P}(G^c \mid \theta \in \Theta_n, \tau_k \leq n) \leq \frac{n+2}{n^2}.
$$

**Bounding the expected additional selection number**   Since we always have $T_\theta^{\text{add}}(n) \leq n$,

$$
\begin{aligned}
\mathbb{E}[T_\theta^{\text{add}}(n) \mid \theta \in \Theta_n, \tau_k \leq n] &= \mathbb{E}[T_\theta^{\text{add}}(n)\mathbb{1}_G \mid \theta \in \Theta_n, \tau_k \leq n] + \mathbb{E}[T_\theta^{\text{add}}(n)\mathbb{1}_{G^c} \mid \theta \in \Theta_n, \tau_k \leq n] \\
&\leq u_{\text{single}} + n\mathbb{P}(G^c \mid \theta \in \Theta_n, \tau_k \leq n) \\
&\leq u_{\text{single}} + n(n+2)/n^2 \\
&\leq \frac{64\sigma^2}{\gamma^2}\log(n) + 3.
\end{aligned}
$$

So we finish the proof of Theorem 3.

### B.3. Proof of Theorem 1

By definition,

$$
\mathbb{E}\bigg[ \sum_{\theta \in \Theta_n : \mu(\theta) \leq B-\gamma} T_\theta(n) \bigg] = \sum_{k=1}^{2B/\gamma-2} \mathbb{E}\bigg[ \sum_{\theta \in I_k \cap \Theta_n} T_\theta(n) \bigg].
$$

For each interval, by definition of $\tau_k$ in (1),

$$
\mathbb{E}\left[\sum_{\theta \in I_k \cap \Theta_n} T_\theta(n)\right] = \mathbb{E}\left[\sum_{\theta \in I_k \cap \Theta_n} T_\theta(n) \mid \tau_k \leq n\right]\mathbb{P}[\tau_k \leq n] + \mathbb{E}\left[\sum_{\theta \in I_k \cap \Theta_n} T_\theta(n) \mid \tau_k > n\right]\mathbb{P}[\tau_k > n]
$$

$$
\leq \mathbb{E}\left[u_{\text{interval}} + \sum_{\theta \in I_k \cap \Theta_n} T_\theta^{\text{add}}(n) \mid \tau_k \leq n\right]\mathbb{P}[\tau_k \leq n] + u_{\text{interval}} \cdot \mathbb{P}[\tau_k > n]
$$

$$
\leq u_{\text{interval}} + \mathbb{E}\left[\sum_{\theta \in I_k \cap \Theta_n} T_\theta^{\text{add}}(n) \mid \tau_k \leq n\right]. \tag{2}
$$

Since $\theta \notin \Theta_n \Rightarrow T_\theta^{\text{add}}(n) = 0$, we have

$$
\mathbb{E}[T_\theta^{\text{add}}(n) \mid \tau_k \leq n] = \mathbb{E}[T_\theta^{\text{add}}(n) \mid \theta \in \Theta_n, \tau_k \leq n] \cdot \mathbb{P}[\theta \in \Theta_n] + \mathbb{E}[T_\theta^{\text{add}}(n) \mid \theta \notin \Theta_n, \tau_k \leq n] \cdot \mathbb{P}[\theta \notin \Theta_n]
$$

$$
\leq \mathbb{E}[T_\theta^{\text{add}}(n) \mid \theta \in \Theta_n, \tau_k \leq n]. \tag{3}
$$

Therefore,

$$
\mathbb{E}\left[\sum_{\theta \in \Theta_n : \mu(\theta) \leq B - \gamma} T_\theta(n)\right] = \sum_{k=1}^{2B/\gamma - 2} \mathbb{E}\left[\sum_{\theta \in I_k \cap \Theta_n} T_\theta(n)\right]
$$

$$
\overset{(2)}{\leq} \sum_{k=1}^{2B/\gamma - 2}\left(u_{\text{interval}} + \mathbb{E}\left[\sum_{\theta \in I_k \cap \Theta_n} T_\theta^{\text{add}}(n) \mid \tau_k \leq n\right]\right)
$$

$$
\leq 2B/\gamma \cdot u_{\text{interval}} + \mathbb{E}\left[\sum_{\theta \in \Theta_n : \mu(\theta) \leq B - \gamma} T_\theta^{\text{add}}(n) \mid \tau_k \leq n\right]
$$

$$
\overset{(3)}{\leq} 2B/\gamma \cdot u_{\text{interval}} + \mathbb{E}\left[\sum_{\theta \in \Theta_n : \mu(\theta) \leq B - \gamma} T_\theta^{\text{add}}(n) \mid \theta \in \Theta_n, \tau_k \leq n\right]
$$

$$
= 2B/\gamma \cdot u_{\text{interval}} + \mathbb{E}\left[\sum_{\theta \in \Theta_n : \mu(\theta) \leq B - \gamma} \mathbb{E}\left[T_\theta^{\text{add}}(n) \mid \theta \in \Theta_n, \tau_k \leq n\right] \mid \Theta_n\right]
$$

$$
\leq 2B/\gamma \cdot u_{\text{interval}} + N_\varepsilon \cdot (u_{\text{single}} + 3),
$$

where the last step is from Theorem 3 and the fact that the cardinality of $\Theta_n$ is bounded by $N_\varepsilon$.

### B.4. Proof of Theorem 2

For simplicity, we assume that $B$ is divisible by $\gamma$. Let $R_t$ denote the reward of the program proposed at step $t$. This induces a probability measure on the sequence $R = (R_1, R_2, \ldots, R_t, \ldots)$.

**Sequential Updates.** Suppose the sequence $R = (R_1, R_2, \ldots, R_t, \ldots)$ is generated by a sequential updating algorithm following (4):

$$
\theta'_t \sim \Pi(\cdot \mid \mathcal{C}_{\theta_{t-1}}). \tag{4}
$$

Then, by Assumption 1, we have

$$
\mathbb{P}[R_{t+1} > R_t + \gamma] \geq \delta_0.
$$

Intuitively, in the worst case, a sequential updating algorithm must make $N := B/\gamma$ consecutive improvements to reach a near-optimal program. We define the stopping time $\tau$ as the first step achieving a reward in $(B - \gamma, B]$, and $\tau_1$ as the first step where an improvement larger than $\gamma$ has occurred for $N$ consecutive steps:

$$
\tau := \min\{t : R_t > B - \gamma\},
$$

$$
\tau_1 := \min\{t : R_t - R_{t-1} > \gamma, \ldots, R_{t-N+1} - R_{t-N} > \gamma\}.
$$

For any given reward sequence $R$, the set of indices satisfying the condition for $\tau_1$ is a subset of those satisfying the condition for $\tau$; therefore, $\tau \leq \tau_1$.

In the worst case, we assume $\mathbb{P}[R_s - R_{s-1} > \gamma] = \delta_0$ for all $s$. Define $\tau_1^{(n)} = \min\{t : R_t - R_{t-1} > \gamma, \ldots, R_{t-n+1} - R_{t-n} > \gamma\}$ as the first time $n$ consecutive improvements occur, where $\tau_1 = \tau_1^{(N)}$.

By conditioning on the outcome after achieving $n - 1$ consecutive improvements, we have:

$$\mathbb{E}[\tau_1^{(n)}] = \delta_0(\mathbb{E}[\tau_1^{(n-1)}] + 1) + (1 - \delta_0)(\mathbb{E}[\tau_1^{(n-1)}] + 1 + \mathbb{E}[\tau_1^{(n)}]),$$

where the second term accounts for the streak being broken, requiring a full restart. Simplifying this expression yields:

$$\mathbb{E}[\tau_1^{(n)}] = \frac{\mathbb{E}[\tau_1^{(n-1)}] + 1}{\delta_0}.$$

Using the base case $\mathbb{E}[\tau_1^{(0)}] = 0$, the closed-form solution for this recurrence is:

$$\mathbb{E}[\tau_1] = \frac{\delta_0^{-N} - 1}{1 - \delta_0}.$$

This result establishes the $\mathcal{O}(\delta_0^{-N}) = \mathcal{O}(\delta_0^{-B/\gamma})$ complexity for the sequential updating algorithm.

**POLCA.** Suppose the sequence $R = (R_1, R_2, \ldots, R_t, \ldots)$ is generated by POLCA using the updating rule (5):

$$\theta_t' \sim \Pi(\cdot \mid \mathcal{C}_{\tilde{\theta}_t}), \quad \text{where} \quad \tilde{\theta}_t := \underset{\theta \in \Theta_t}{\operatorname{argmax}} \, \mu(\theta). \tag{5}$$

Let $\tau$ be the first step reaching a reward in $(B - \gamma, B]$. We define $\tau_2$ as the first step where $N$ total (not necessarily consecutive) improvements of size at least $\gamma$ have been observed:

$$\tau = \min\{t : R_t > B - \gamma\},$$

$$\tau_2 = \min\left\{t : \sum_{s=1}^{t} \mathbb{I}(R_s > \max_{j<s} R_j + \gamma) \geq N\right\}.$$

Under the POLCA rule, each proposal is conditioned on the best observation found so far. Therefore, reaching $N$ cumulative improvements is sufficient to achieve a near-optimal reward, so $\tau \leq \tau_2$.

The time between each improvement is an independent geometric random variable with mean $1/\delta_0$. By the linearity of expectation, we have:

$$\mathbb{E}[\tau_2] = \sum_{i=1}^{N} \frac{1}{\delta_0} = \frac{N}{\delta_0}.$$

Substituting $N = B/\gamma$ yields $\mathbb{E}[\tau] \leq B/(\gamma\delta_0)$, completing the proof of the efficiency of POLCA.

**Discussion** If we do not assume a lower bound for $\mu$, the complexity of POLCA remains $\mathcal{O}(N/\delta_0)$, whereas the sequential updating algorithm may fail to converge. In the absence of a lower bound, making $N$ consecutive improvements is no longer sufficient to reach a near-optimal program if a single "bad" proposal at step $t$ results in a reward $R_t \ll R_0$. Specifically, if $\mu$ is not lower-bounded (e.g., $R \in (-\infty, B]$), a single failure with probability $1 - \delta_0$ could yield an arbitrarily small reward, requiring significantly more than $N$ subsequent improvements to recover. In the worst case, such as a process where a failure results in $R_t = -\infty$, the expected number of steps for a sequential updating algorithm to reach $B - \gamma$ becomes infinite. In contrast, POLCA is robust to such "catastrophic" proposals because it always updates from the historical maximum $\tilde{\theta}_t$, ensuring the baseline reward is non-decreasing.

## C. Implementation Details of POLCA

In this section we elaborate the detailed logic of components of POLCA.

**Program Execution and Evaluation** The EVALUATE function serves as a general-purpose utility to execute and evaluate a subset of candidates $\tilde{\Theta}$ on task minibatch $\mathcal{B}$ and generate the rollout set $\mathcal{S}$. For each task input $x_i$, the program $P_\theta$ produces an output $y_i \sim P_\theta(x_i)$, evaluated by the Guide $\mathcal{G}$ to obtain rewards $r_i$ and feedback $f_i$. To maximize throughput, the process is executed asynchronously, treating each program-task interaction as an independent parallel task. These are aggregated into rollout tuples $s = (\theta, \omega, x, y, r, f)$, forming the set $\mathcal{S} = \bigcup_{\theta \in \tilde{\Theta}} \mathcal{S}_\theta$. The asynchronous EVALUATE procedure is detailed in Algorithm 2.

---

**Algorithm 2** EVALUATE – Asynchronous program execution and evaluation

---

**Require:**
    **Inputs:** Candidate agents $\tilde{\Theta}$, Minibatch $\mathcal{B}$, Guide $\mathcal{G} = (\mathcal{G}_r, \mathcal{G}_f)$
**Ensure:** Aggregate rollout set $\mathcal{S}$
1:   $\mathcal{S} \leftarrow \emptyset$
2:   $\mathcal{T} \leftarrow \emptyset$                                          ▷ Global task queue for concurrent execution
3:   **for** each program $\theta \in \tilde{\Theta}$ **do**
4:      **for** each $(\omega, x) \in \mathcal{B}$ **do**
5:          $\mathcal{T} \leftarrow \mathcal{T} \cup \{(\theta, x, \omega)\}$                 ▷ Queue program-task pairs for parallel dispatch
6:      **end for**
7:   **end for**
8:   **for** each $(\theta, x, \omega) \in \mathcal{T}$ **in parallel do**                   ▷ Fully asynchronous thread execution
9:      $y \sim P_\theta(x)$                                        ▷ Program execution
10:     $r \sim \mathcal{G}_r(\omega, x, y)$                            ▷ Numerical reward evaluation
11:     $f \sim \mathcal{G}_f(\omega, x, y)$                        ▷ Textual feedback generation
12:     $s \leftarrow (\theta, \omega, x, y, r, f)$                    ▷ Construct rollout tuple
13:     **lock** $\mathcal{S} \leftarrow \mathcal{S} \cup \{s\}$
14:   **end for**
15:   **wait** for all threads in $\mathcal{T}$ to return
16:   **return** $\mathcal{S}$

---

**Context Construction and Program Proposal.** In the PROPOSEPROGRAMS function, the optimizer $\mathcal{O}$ utilizes both the local data $\mathcal{S}$ and global context from $\mathcal{Q}$ to generate a set of new program parameters $\Theta_{\text{new}}$, which are sampled from the proposal distribution $\Pi(\cdot \mid \mathcal{C})$.

Like the role of the gradient in numerical optimization algorithms, our proposal mechanism is mainly based on local revision and improvement. After evaluating the improving programs $\Theta_{\text{explore}}$, POLCA constructs local context $\mathcal{S} = \bigsqcup_{\theta \in \Theta_{\text{explore}}} \mathcal{S}_\theta$. To inject more information from $\mathcal{Q}$, we also utilize an external LLM, called *Summarizer*, to construct a population-level context $c_{\text{history}}$ by distilling the aggregate trajectory[9] history $\mathcal{H} = \bigcup_{\theta \in \mathcal{Q}} \mathcal{H}_\theta$. The Summarizer partitions this history into performance-based subsets: successes $\mathcal{H}_\theta^+ = \{(\theta, \omega, x, y, r, f) \in \mathcal{H}_\theta : r > \tau\}$ and failures $\mathcal{H}_\theta^- = \{(\theta, \omega, x, y, r, f) \in \mathcal{H}_\theta : r \leq \tau\}$ based on a reward threshold $\tau$. The threshold $\tau$ is not a tuned hyperparameter in our experiments: across all four benchmarks ($\tau$-bench, HotpotQA, VeriBench, and KernelBench), the partition $\mathcal{H}_\theta^+ / \mathcal{H}_\theta^-$ corresponds to a natural binary distinction—successful vs. failed trajectories for $\tau$-bench/HotpotQA, and compilable+correct vs. not for VeriBench/KernelBench—and $\tau = 0$ recovers exactly this distinction, so there is nothing to tune and no sensitivity to the threshold value. For more general continuous-reward domains where no such natural cutoff exists, $\tau$ can be set to a robust statistic of the observed rewards (e.g. the median of $\mathcal{H}_\theta$) to ensure a balanced partition; we expect this rule to remain robust without per-task tuning, though we do not study it empirically here. By identifying systematic patterns across these partitions—such as instruction formats unique to high-scoring programs—the Summarizer compresses these insights into natural language *meta-gradients*. To maintain a representative view while adhering to context limits, we employ a *Contrastive Sampling* strategy, providing the LLM with program parameters alongside paired representative trajectories (one $r > \tau$ and one $r \leq \tau$). This enables the LLM to perform cross-program error correction, providing a stable historical direction analogous to *Gradient Descent with Momentum* (Cui et al., 2024). The process is governed by a structured prompt template utilizing XML-style tags to separate internal reasoning from actionable guidance, as detailed below.

---

[9]We also refer to evaluation data of the form $(\theta, \omega, x, y, r, f)$ as an evaluation trajectory.

---

**Summarizer Prompt Template**

**System:** You are an expert at analyzing program behavior patterns and providing actionable guidance for parameter optimization.

**User:** Analyze the following program rollout trajectories and extract insights for optimization. For each program, a successful and a failed trajectory are provided for contrastive analysis.
**Trajectories:**
$\{history\_trajectories\}$

Provide your analysis in XML format:
- `<reasoning>` Analyze the key patterns and strategies that led to success or failure in these trajectories. `</reasoning>`
- `<summary>` Concrete recommendations for improving output quality based on successful or failed patterns observed. `</summary>`

---

Then we combine local rollouts with $c_{\text{history}}$ summarized from $\mathcal{Q}$ to construct the context for $\mathcal{O}$. For each $\theta \in \Theta_{\text{explore}}$, we construct the context as $\mathcal{C}_\theta = \{(\theta, x_i, y_i, r_i, f_i, c_{\text{history}})\}_{i=1}^B$ and invoke the optimizer $\mathcal{O}$ to get $\theta' \sim \Pi(\cdot \mid \{(\theta, x_i, y_i, r_i, f_i, c_{\text{history}})\}_{i=1}^B)$. We collect the proposed program in $\Theta_{\text{raw}}$. The asynchronous PROPOSEPROGRAMS subroutine is detailed in Algorithm 3.

---

**Algorithm 3** PROPOSEPROGRAMS – Fully asynchronous program generation

---

**Require:**
    **Inputs:** Aggregate rollouts $\mathcal{S} = \bigcup_\theta \mathcal{S}_\theta$, priority queue $\mathcal{Q}$, optimizer $\mathcal{O}$
**Ensure:** Set of new program parameters $\Theta_{\text{raw}}$
 1: $\Theta_{\text{raw}} \leftarrow \emptyset$
 2: $\mathcal{U} \leftarrow \emptyset$                                         ▷ Global task queue for parallel execution
 3: Summarize $\mathcal{Q}$ to get global history context $c_{\text{history}}$
 4: **for** each $\theta \in \Theta_{\text{explore}}$ **do**
 5:     Extract the rollouts $\mathcal{S}_\theta = \{(\theta, \omega_i, x_i, y_i, r_i, f_i)\}_{i=1}^{|\mathcal{B}|}$
        to Construct the context $\mathcal{C}_\theta = \{(\theta, x_i, y_i, r_i, f_i, c_{\text{history}})\}_{i=1}^{|\mathcal{B}|}$
 6:     $\mathcal{U} \leftarrow \mathcal{U} \cup \{\mathcal{C}_\theta\}$                               ▷ Queue task for parallel dispatch
 7: **end for**
 8: **for** each $\mathcal{C}_\theta \in \mathcal{U}$ **in parallel do**              ▷ Massively parallel asynchronous calls
 9:     $\theta' \sim \Pi(\cdot \mid \mathcal{C}_\theta)$
10:     **lock** $\Theta_{\text{raw}} \leftarrow \Theta_{\text{raw}} \cup \{\theta'\}$
11: **end for**
12: **wait** for all threads in $\mathcal{U}$ to return
13: **return** $\Theta_{\text{raw}}$

---

**Semantic Filter based on $\varepsilon$-Net Design**    To enhance optimization efficiency, the SEMANTICFILTER subroutine leverages semantic similarity among programs to prune redundant proposals and maintain diversity. Given a set of raw candidates $\Theta_{\text{raw}}$, we construct a filtered set $\Theta_{\text{new}}$ using a farthest-first traversal strategy to form an $\varepsilon$-Net. Starting with $\Theta_{\text{new}} \leftarrow \emptyset$ and a program pool $\Theta_{\text{remaining}} \leftarrow \Theta_{\text{raw}}$, we iteratively: (1) for each $\theta \in \Theta_{\text{remaining}}$, compute its distance to the current population of validated and newly selected agents, $d(\theta) = \min_{\theta' \in \mathcal{Q} \cup \Theta_{\text{new}}} \tilde{d}(\theta, \theta')$; (2) identify the candidate $\theta^* = \arg\max_{\theta \in \Theta_{\text{remaining}}} d(\theta)$ with the maximum distance $d_{\max} = d(\theta^*)$; and (3) if $d_{\max} > \varepsilon$, transfer $\theta^*$ from $\Theta_{\text{remaining}}$ to $\Theta_{\text{new}}$, otherwise the process terminates. This greedy selection strategy ensures the expanded agent set maintains a diversity threshold: for any distinct pair $\theta, \theta' \in \mathcal{Q} \cup \Theta_{\text{new}}$, the semantic distance satisfies $\tilde{d}(\theta, \theta') > \varepsilon$. Implementation details are provided in Algorithm 4.

## D. Instantiating Classical Search Algorithms

POLCA is universal because modifying the priority function $p_{\text{explore}} : \Theta \rightarrow \mathbb{R}$ is sufficient to mimic many classical search paradigms. Changing $p_{\text{explore}}(\cdot)$ leaves the rest of the algorithm untouched, making it easy to implement different designs on the same problem instance.

**Sequential search (Iterative refinement).**    The simplest strategy follows a depth-first trajectory through the search space. At each iteration, only the most recently proposed program is selected for further evaluation and refinement. This behavior

---

**Algorithm 4** SEMANTICFILTER – Semantic-based pruning for candidate diversity

---

**Require:**
    **Inputs:** Raw candidate set $\Theta_{\text{raw}}$, priority queue $\mathcal{Q}$
    **Hyperparameters:** Diversity threshold $\varepsilon$, semantic distance metric $\tilde{d}(\cdot, \cdot)$
**Ensure:** Filtered set $\Theta_{\text{new}}$ s.t. $\forall\, \theta_i, \theta_j \in (\Theta_{\text{new}} \cup \mathcal{Q}), i \neq j \implies \tilde{d}(\theta_i, \theta_j) \geq \varepsilon$

1:
2:   $\Theta_{\text{new}} \leftarrow \emptyset$
3:   $\Theta_{\text{remaining}} \leftarrow \Theta_{\text{raw}}$                                      ▷ Initialize pool of remaining candidates
4:
5: **while** $\Theta_{\text{remaining}} \neq \emptyset$ **do**
6:                                          ▷ Find candidate with maximum distance to the existing population
7:      $\theta^* \leftarrow \arg\max_{\theta \in \Theta_{\text{rem}}} \left\{ \min_{\theta' \in \mathcal{Q} \cup \Theta_{\text{new}}} \tilde{d}(\theta, \theta') \right\}$
8:      $\delta_{\max} \leftarrow \min_{\theta' \in \mathcal{Q} \cup \Theta_{\text{new}}} \tilde{d}(\theta^*, \theta')$
9:
10:      **if** $\delta_{\max} \geq \varepsilon$ **then**
11:          $\Theta_{\text{new}} \leftarrow \Theta_{\text{new}} \cup \{\theta^*\}$
12:          $\Theta_{\text{remaining}} \leftarrow \Theta_{\text{remaining}} \setminus \{\theta^*\}$
13:      **else**
14:          **break**                     ▷ Termination: all remaining candidates are semantically redundant
15:      **end if**
16: **end while**
17:
18: **return** $\Theta_{\text{new}}$

---

is enforced by setting $p_{\text{explore}}(\theta) = t_\theta$, where $t_\theta$ is the creation timestamp of agent $\theta$. By using this *Last-In, First-Out* (LIFO) ordering and restricting the exploration budget to $k = 1$, the algorithm collapses into a sequential refinement process that ignores the broader population in favor of local, iterative improvements.

**Beam search.** Sequential search is prone to local traps. Beam search improves robustness by maintaining a population of $k$ active *beams*; at each iteration, these beams produce new programs that are validated, with only the top-$k$ candidates surviving based on their initial scores.

In our framework, this is realized by allowing $\mathcal{O}$ to propose multiple new programs $\theta'$, and assigning $p_{\text{explore}}(\theta') = \bar{r}(\theta')$ for newly proposed programs—where the average is calculated using only the validation data from the current iteration—and setting $p_{\text{explore}}(\theta) = -\infty$ for all old programs in memory. This ensures the priority queue retains only the most recent high-performing generation while pruning older branches, effectively emulating the breadth-first expansion of classical beam search. This approach might be efficient in deterministic settings; however, in the presence of stochasticity, discarding historical evaluations and evaluating each program only once may lead to suboptimal results.

**Upper Confidence Bound (UCB)** In finite-action best-arm-identification bandit theory, greedy exploration is not provably optimal. To address this, more refined strategies such as the Upper Confidence Bound (UCB) approach can be employed to provide an explicit incentive for collecting data on uncertain candidates. UCB incorporates an uncertainty bonus into the priority calculation, specifically, at iteration $t$ we can calculate the priority score for each program as:

$$p_{\text{explore}}(\theta) = \widehat{\mu}_{\theta, T_\theta(t)} + \beta \sqrt{\frac{\log(n)}{T_\theta(t)}},$$

where $T_\theta(t)$ is the number of reward observations of program $\theta$, $\widehat{\mu}_{\theta,s}$ is the empirical mean of program $\theta$ with the first $s$ reward observations, $n = \sum_{\theta' \in \mathcal{Q}} T_{\theta'}(t)$ is the total rollout budget, and $\beta > 0$ controls the exploration-exploitation tradeoff. The bonus increases for programs with small $T_\theta(t)$, guiding exploration toward under-sampled regions and preventing premature convergence. In Section 4, we prove if $\beta$ is selected as a certain number related to the randomness of the system, under some assumptions, a simplified version of POLCA could converge to programs that can not be further improved by the optimizer.

While our current framework utilizes the empirical mean as a robust starting point for stochastic generative optimization, integrating such carefully designed priority functions represents a promising direction for future work to further enhance search performance and accelerate the identification of the global optimum.

# E. Detailed Experiments

## E.1. Baselines

**DSPy** DSPy (Khattab et al., 2023) is a declarative framework designed for modularizing and optimizing Large Language Model (LLM) pipelines. It provides a structured approach to prompt engineering by treating prompts as learnable parameters within a programmatic workflow. It can also be adapted to optimize general-purpose string-based programs with well-defined rewards and feedback. In our experiments, we use a `dspy.ChainOfThought` module to implement a sequential revision search algorithm, which always takes the current parameter with its score and feedback to propose a new parameter at each step. In the following we use **DSPy** to denote such a search algorithm.

**GEPA** GEPA (Genetic-Pareto Prompt Optimizer) is a reflective prompt optimization algorithm recently integrated into the DSPy ecosystem (Agrawal et al., 2025). We apply it here to more diverse generative optimization problems beyond prompt tuning by simply treating the optimizable parameter as a string. It maintains a *Pareto frontier* of parameters to track non-dominated solutions across different training instances, leveraging natural language reflection to iteratively improve parameters through trial and error.

**OpenEvolve** OpenEvolve (Sharma, 2025) is an open-source implementation of AlphaEvolve (Novikov et al., 2025), an autonomous evolutionary pipeline for algorithmic discovery. We could also utilize it to handle diverse generative optimization problems beyond code evolution. It utilizes the MAP-Elites algorithm and island-based search to manage a population of diverse, high-performing parameters through iterative evaluation and selection.

## E.2. Task Formulation and Implementation Details

In this section, we describe the formulation of generative optimization tasks across various domains and specify the configuration for all evaluated search algorithms.

$\tau$**-bench** $\tau$-bench is a benchmark designed to evaluate multi-turn agents on their ability to interact with human users, adhere strictly to domain-specific policies, and execute tools to resolve complex queries. The environment provides a sparse, binary reward $r \in \{0, 1\}$ for each program-task execution, indicating whether the user's request was successfully resolved. We utilize the first 10 tasks from the retail domain of $\tau$-bench for optimization and the remaining 105 tasks held out to test[10] the generalization of the optimized agent. The base agent (program) provided by the benchmark is parameterized via a string variable, `additional_instructions`, which is appended to the original system prompt. Search algorithms learn optimized versions of `additional_instructions` through trial and error by reflecting on the accumulated conversation history.

For our experiments we use `gemini-2.0-flash` as the backbone model of agents, simulated users in $\tau$-bench and optimizers in search algorithms, `gemini/text-embedding-004` as the embedding model for POLCA. We do external tests to measure the performance of trained agent instances from different search algorithms. The test score for a specific agent instance is computed by running 10 trials per task and calculating the average pass rate across all tasks and trials[11]. For all the search algorithms, we set the maximum number of parallel evaluations in one evaluation step to be 10, meaning that when running the algorithm, at most 10 evaluations could be done in parallel at the same time. For POLCA, we set `num_candidates = 5`, `batch_size = 2`, and `num_batches = 1`. We select the candidates with the highest mean scores for external tests. For OpenEvolve, we construct an evaluator that runs the proposed agent instance on all 10 tasks once and use the pass rate as the score. We set the `parallel_evaluations = 10` and `max_workers = 1`, which means OpenEvolve could evaluate an agent instance on 10 tasks in parallel but cannot evaluate multiple agent instances in parallel. We choose `num_islands = 3`, `migration_interval = 20`, and `migration_rate = 0.1` to perform island-based evolution. We set `num_top_programs = 3` and `num_diverse_programs = 2` as the programs to include in the prompt. We pick the candidate with the highest pass rate for external tests. For GEPA, we also choose `batch_size = 2`, and use all 10 tasks as the

---

[10]These tasks are exclusively used for the final evaluation presented in Table 1. Other external tests involve evaluating the first 10 tasks multiple times to obtain a nearly deterministic score for the trained agents.

[11]Here test score of each agent instance is an average of 100 trials, which serves as an accurate measure of performance.

validation dataset. In the training process it will maintain a pareto-frontier of non-dominated agent instances for 10 tasks. For external test, we implement two ways to pick the best agent instances from the training of GEPA. Specifically, we evaluate two selection methods: *1) GEPA (most freq)*, which selects the candidate appearing most frequently in the Pareto frontier—this corresponds to the agent achieving the highest pass rate over 10 tasks—and *2) GEPA (sample freq)*[12], which performs weighted selection based on candidate appearance frequency in the pareto-frontier. Results are averaged over 6 random seeds, showing the mean and standard error for independent runs in Figure 2.

**HotpotQA**    HotpotQA (Yang et al., 2018) is a multi-hop question answering dataset where each task requires reasoning across multiple context paragraphs to produce a short answer. We use the distractor setting of HotpotQA, taking the first 100 examples from the validation split as our benchmark. Each example consists of a question, 10 context paragraphs (of which 2–3 are relevant and the rest are distractors), and a ground-truth answer. We use `gemini-2.5-flash-lite` as the backbone model for task execution, and `gemini-embedding-001` as the embedding model for POLCA. Correctness is determined by case-insensitive exact match or substring containment after stripping trailing punctuation, yielding a binary reward of 0 or 1 per task. Test scores are computed by evaluating each candidate prompt on all 100 tasks with 5 independent repetitions per task, and we report the average accuracy across repetitions.

For GEPA, we use a reflection minibatch size of 10 with a budget of 2,000 metric calls per run. For OpenEvolve, we run 50 iterations with cascade evaluation (a 20-sample first stage with a $\geq 50\%$ threshold gating a full 100-sample second stage). For POLCA, we set $\varepsilon = 0.1$, num_candidates = 5, batch_size = 2, and num_batches = 1. All algorithms use `gemini-2.5-flash-lite` for meta-optimization (reflection/proposal generation). We conduct 3 independent runs per algorithm and report the mean and standard error.

**VeriBench (3-step evaluation)**    VeriBench (Miranda et al., 2025) is a challenging domain for current LLMs due to their limited domain knowledge of Lean 4 programming. It evaluates the capability of LLMs to translate Python programs into verifiable Lean 4 code. Each task contains one Python program and a golden Lean 4 program. LLMs are prompted to translate the Python program into a compilable and semantically correct Lean 4 program. We formalize this problem by treating the entire translated Lean 4 program as the parameter, i.e., $P_\theta(x) \equiv \theta$, resulting in a fully deterministic program execution.

We select the `easy_set` (41 tasks) from VeriBench and optimize for each individual task, utilizing a sequential three-step evaluation process for each Lean 4 program. For each candidate, the program is first processed by a Lean 4 compiler within a Lean 4 RL environment accessed via PyPantograph (Aniva et al., 2025). The compiler returns a binary score; if compilation fails, the compiler error is captured as textual feedback. A candidate only proceeds to the subsequent stage if it successfully passes the current check. If compilation succeeds, the program is evaluated against several unit tests to determine if the Lean 4 translation is semantically correct and functionally equivalent to the original Python program. Finally, only programs that pass both the compiler and the unit tests are assessed by an LLM judge. The judge compares the translated Lean 4 program with the ground-truth implementation, assigning a score from 0 to 30. Consequently, programs that fail at earlier stages receive a score of zero. This LLM-based score is subsequently normalized to $[0, 1]$ for use in the optimization process. Since this three-step evaluation is progressive, we design a numerical reward signal to determine the final performance of the proposed agent instances. The reward $r \in [0, 1]$ is defined as:

$$r = 0.3 \cdot \mathbb{1}_{\text{Compilation}} + 0.3 \cdot \mathbb{1}_{\text{Unit tests}} + 0.4 \cdot r_{\text{LLM}}.$$

While the first two phases are fully deterministic, the final component introduces stochasticity through the score and feedback provided by the LLM judge.

We use `claude-3.5-sonnet` as the backbone for search algorithms and LLM judge, and `gemini/text-embedding-004` as the embedding model, evaluating DSPy, GEPA, OpenEvolve, and POLCA. For all the search algorithms, we set the maximum number of parallel evaluations in one evaluation step to be 5, meaning that when running the algorithm, at most 5 evaluations could be done in parallel at the same time. For GEPA, we set batch_size = 1 and use the single task under optimization as the validation dataset. For OpenEvolve, we set num_islands = 1 and num_workers = 5 for simplicity. For our POLCA variants, we set the exploration budget $k = 5$ and the diversity threshold $\varepsilon = 0.02$.

We compare algorithms by measuring the pass rate achieved over the tasks within a fixed budget. For each algorithm, we perform the search across all 41 tasks independently, with a maximum budget of 50 compiler calls per task. We calculate the

---

[12]This metric is utilized to select the programs for exploration within GEPA.

average score over all tasks at each step and for each number of metric calls. To ensure statistical reliability, we repeat the experiments three times and report the mean and standard error across these runs.

**VeriBench (Compilation)**    We utilize the complete VeriBench dataset, consisting of all 140 tasks, focusing specifically on the compilation stage. This remains a challenging domain given the limited Lean 4 programming knowledge inherent in current LLMs. For this analysis, we define the reward as a binary indicator of compilation success, $r = \mathbb{1}_{\text{Compilation}} \in \{0, 1\}$, while all other experimental settings remain unchanged. We run each search algorithm on the 140 tasks independently and repeat the experiments three times.

**KernelBench**    CUDA kernel optimization is also a popular domain in generative optimization problems. In this part, we pick 16 matrix multiplication tasks from KernelBench (level 1) (Ouyang et al., 2025), which appear simple but remain challenging. As mentioned in Yan et al. (2026), these tasks are already highly optimized in PyTorch, making it difficult to achieve further speedups.

We use the `claude-3.7-sonnet` model and the `gemini-embedding-001` embedding model. For GEPA, we set `batch_size` $= 1$ and use the single task under optimization as the validation dataset. For OpenEvolve, we set `num_islands` $= 1$ and `num_workers` $= 5$ for simplicity. For our POLCA variants, we set the exploration budget $k = 5$ and the diversity threshold $\varepsilon = 0.02$. The kernel evaluation is executed on an `L40S` GPU, with each evaluation result being an average of five repeated executions. We continue to perform per-task optimization; as the programs being optimized are the kernel programs themselves, there is no stochasticity in execution or minibatch sampling. Although minor noise in code execution speed exists, we have confirmed that this variance is negligible for the optimization process. We utilize the $\text{fast}_p$ score (Ouyang et al., 2025) defined as:

$$\text{fast}_p = \frac{1}{N} \sum_{i=1}^{N} \mathbb{1}(\text{correct}_i \wedge \{\text{speedup}_i > p\}),$$

where $p$ is the speedup threshold relative to the PyTorch baseline. This metric measures the proportion of tasks for which the algorithm proposes a correct CUDA program with a speedup exceeding $p$. Here, $N = 16$ for our selected tasks.

### E.3. Evaluation Metrics

The primary bottleneck in generative optimization varies significantly across domains. For most stochastic problems addressed in this paper, evaluating a single program is expensive due to inherent noise and the computational cost of the evaluation function. While the main paper focuses on evaluation steps, we provide a more comprehensive analysis here by considering additional dimensions of complexity.

**Number of metric calls (Computational complexity)**    Evaluating proposed programs is resource-intensive. By fixing the total number of metric calls, we compare the efficiency of different search algorithms within a strict computational budget.

**Number of evaluation steps (Time complexity)**    We define an *evaluation step* as a unit where all constituent metric calls are executed in parallel. This metric measures the number of sequential operations required, serving as a proxy for wall-clock time.

However, in domains where evaluation is relatively inexpensive, the bottleneck shifts to the proposal process. To provide a multidimensional comparison, we introduce:

**Number of proposals (Computational complexity)**    Generating new candidates requires substantial compute. By fixing the number of proposals, we evaluate how effectively each algorithm utilizes its generative budget.

**Number of proposal steps (Time complexity)**    In each *proposal step*, LLM API calls for program generation are parallelized. This metric counts the sequential operations required for generation. To ensure fairness, we impose a maximum number of parallel proposals per step across all algorithms.

For all experiments, we complement the evaluation step analysis with plots based on these additional metrics to provide deeper insights into search efficiency. Results are presented in Figures 5 to 8.

**Discussion** Due to its parallelized batch update design, POLCA consistently outperforms baselines in terms of proposal and evaluation steps. When considering the total computational budget, such as the number of metric calls and proposals, POLCA maintains strong final performance but may lag behind sequential methods in the early stages of search.

On VeriBench, for instance, POLCA outperforms all parallelized baselines in total metric calls, though it remains slightly below sequential DSPy initially. This occurs because batch-oriented algorithms are designed for low latency and utilize less cumulative information per step compared to sequential updates. For example, with a budget of 10 metric calls, DSPy can perform 10 sequential revisions, reaching a search tree depth of 10. Conversely, POLCA may evaluate 5 candidates in parallel over 2 steps. While the computational cost is identical, POLCA significantly reduces the required sequential time.

Similarly, on KernelBench, while POLCA consistently surpasses baselines in terms of time budget, it does not consistently outperform sequential GEPA when measured by total metric calls. This underscores the inherent trade-off in batch design; sequential methods can achieve greater search depth for the same computation budget, whereas POLCA prioritizes minimizing wall-clock time through parallel execution and batch-informed optimization.

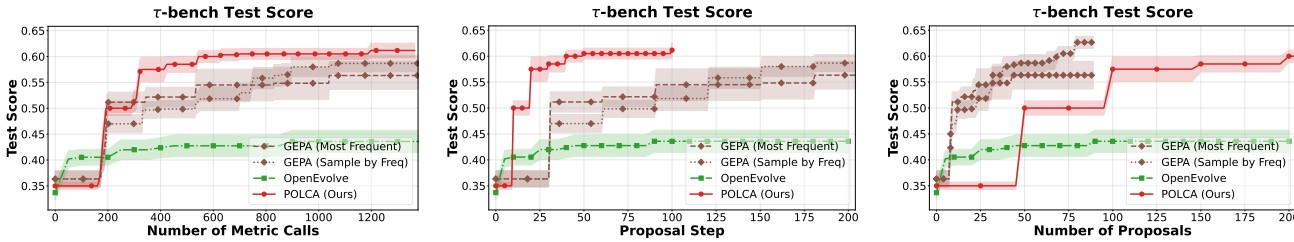

*Figure 5.* $\tau$-bench: performance vs. number of samples (left), proposal steps (middle), and number of proposals (right). Solid curves represent the average highest score attained at each step, while the shaded regions denote the standard error across multiple independent runs (6 seeds).

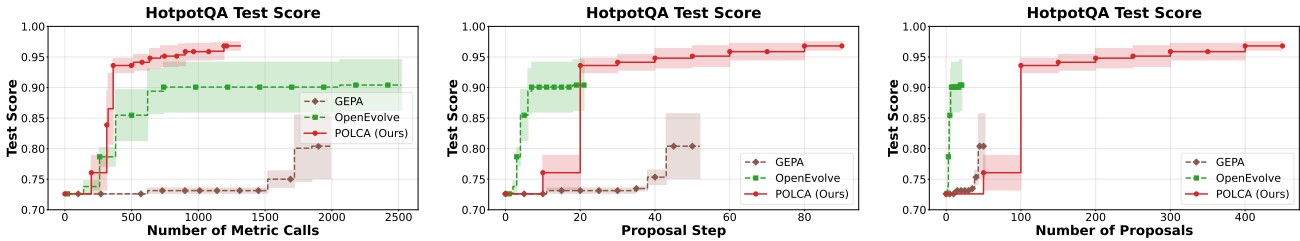

*Figure 6.* HotpotQA: performance vs. number of samples (left), proposal steps (middle), and number of proposals (right). Solid curves represent the average highest score attained at each step, while the shaded regions denote the standard error across multiple independent runs (3 seeds).

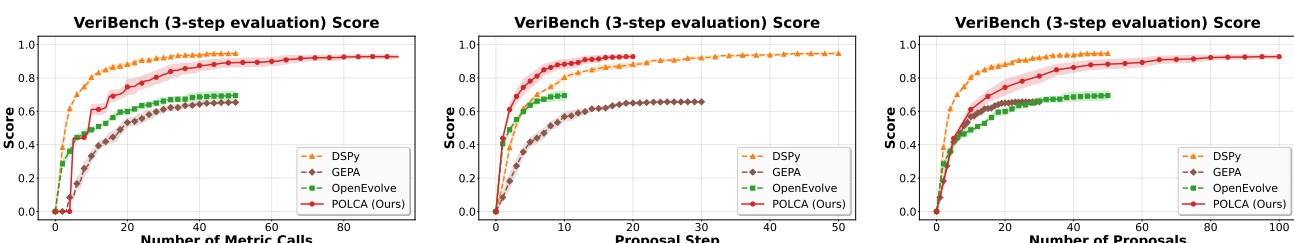

*Figure 7.* VeriBench: performance vs. number of samples (left), proposal steps (middle), and number of proposals (right). Solid curves represent the average highest score attained at each step, while the shaded regions denote the standard error across multiple independent runs (3 seeds).

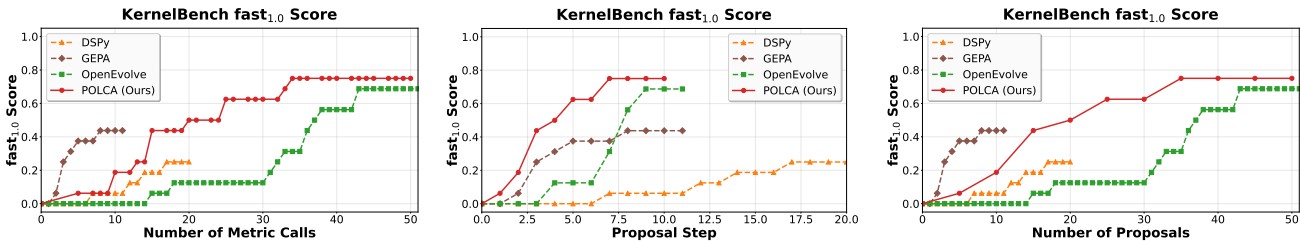

*Figure 8.* KernelBench: performance vs. number of samples (left), proposal steps (middle), and number of proposals (right). Solid curves represent the average highest score attained at each step (1 seed).

### E.4. Compilation Rate Study on VeriBench

We evaluate algorithms on VeriBench (Compilation). Figure 9 compares compilation rates across different evaluation metrics, where POLCA consistently outperforms all baselines.

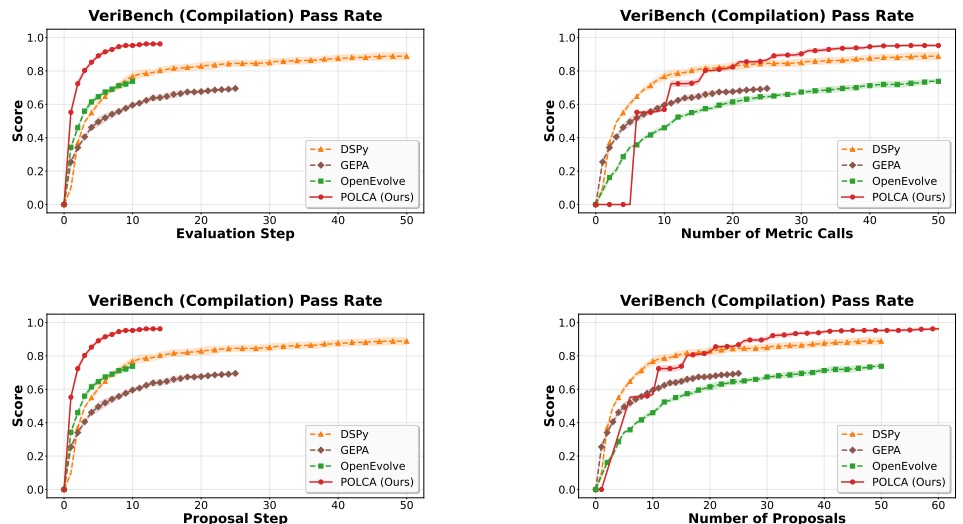

*Figure 9.* VeriBench (Compilation) results: Plots show the compilation rate over 140 tasks relative to number of evaluation steps, metric calls, proposal steps and proposals, respectively. Solid curves represent the average highest score attained at each step, while the shaded regions denote the standard error across multiple independent runs (3 seeds).

### E.5. Fast$_{0.5}$ study on KernelBench

For KernelBench, we also provide a comparison using the pass$_{0.5}$ metric. This metric is significant because PyTorch operators are already highly optimized; therefore, generating custom CUDA kernels from scratch—rather than utilizing pre-wrapped PyTorch functions—that are both correct and reach half the execution speed of the PyTorch implementation represents a meaningful achievement in automated kernel synthesis. Figure 10 shows POLCA consistently outperforms all baselines in terms of evaluation and proposal steps, as well as final performance. While POLCA demonstrates superior efficiency within a fixed time budget, it may not be the most efficient at early stages when compared against computational budgets, such as the total number of metric calls and proposals, particularly against sequential algorithms, for the same reasons discussed in Section E.3.

### E.6. Ablation on UCB Exploration

While our theoretical analysis (Section 4) employs a UCB priority to obtain provable guarantees, throughout the empirical sections we use the empirical mean score $\widehat{\mu}_{\theta, T_{\theta}(t)}$ as the exploration priority. Here we empirically investigate whether replacing the mean with a UCB priority improves the performance of POLCA in practice. Following the form introduced in

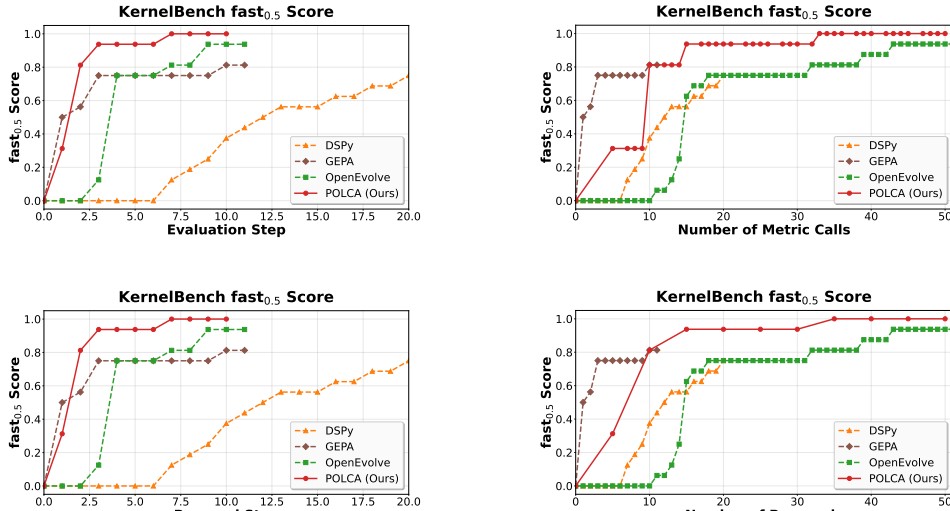

*Figure 10.* Performance comparison across 16 kernel optimization tasks. Plots show the $\text{fast}_{0.5}$ score relative to number of evaluation steps, metric calls, proposal steps, and proposals, respectively. Solid curves represent the average highest score attained at each step (1 seed).

Section D, we consider the tunable priority

$$p_{\text{explore}}(\theta) = \widehat{\mu}_{\theta, T_\theta(t)} + \beta \sqrt{\frac{\log(n)}{T_\theta(t)}}, \tag{6}$$

where $T_\theta(t)$ is the number of reward observations of program $\theta$, $n$ is the total number of rollouts, and $\beta \geq 0$ controls the strength of the exploration bonus. Note that $\beta = 0$ recovers the empirical mean priority used in our default implementation. Setting $\beta = 2\sigma$ matches the theoretical UCB priority in Section 4, which yields the provable guarantee in Theorem 1; however, this requires knowledge of the sub-Gaussian parameter $\sigma$ that characterizes the noise of the reward distribution, which is typically not available a priori. We sweep $\beta \in \{0, 0.05, 0.1, 0.2\}$ on HotpotQA and report the test score against four different evaluation budgets.

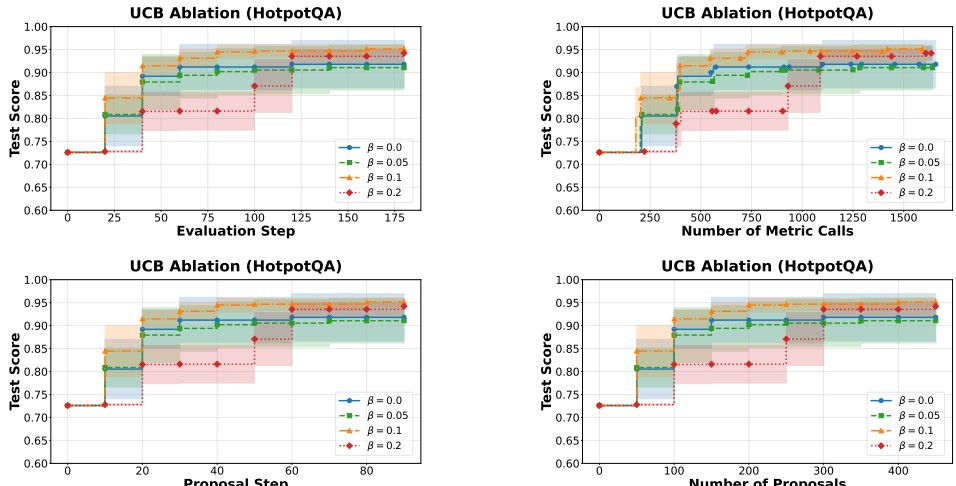

*Figure 11.* Ablation on the UCB exploration coefficient $\beta$ in (6) on HotpotQA. The four panels report test scores versus evaluation steps, metric calls, proposal steps, and number of proposals, respectively. Solid curves represent the average highest score attained at each step, while shaded regions denote the standard error across multiple independent runs (3 seeds).

As shown in Figure 11, a properly tuned UCB bonus can outperform the empirical mean: $\beta = 0.1$ achieves the best

test score across all four budget axes, while $\beta = 0.05$ is nearly indistinguishable from the mean baseline and $\beta = 0.2$ over-weights uncertainty and hurts performance, especially in early optimization. The optimal $\beta$ is thus problem-dependent: the theoretical choice $\beta = 2\sigma$ is provably good but requires knowing the noise level $\sigma$, which is task-specific. Without offline data to estimate $\sigma$ or sweep over $\beta$, a poorly chosen value can be worse than no bonus at all.

In contrast, the empirical mean priority ($\beta = 0$) is already competitive with the best UCB setting and clearly outperforms the worst, while requiring no tuning. We therefore adopt the empirical mean as the default priority in our main experiments, and reserve UCB for the theoretical analysis in Section 4, where the bonus is essential for provable systematic exploration.

### E.7. More Ablation Results

In this section, we provide additional results from our ablation study. Figure 12 shows the ablation on the $\varepsilon$-Net and Summarizer features of POLCA, while Figure 13 presents the ablation study on $\varepsilon$ values.

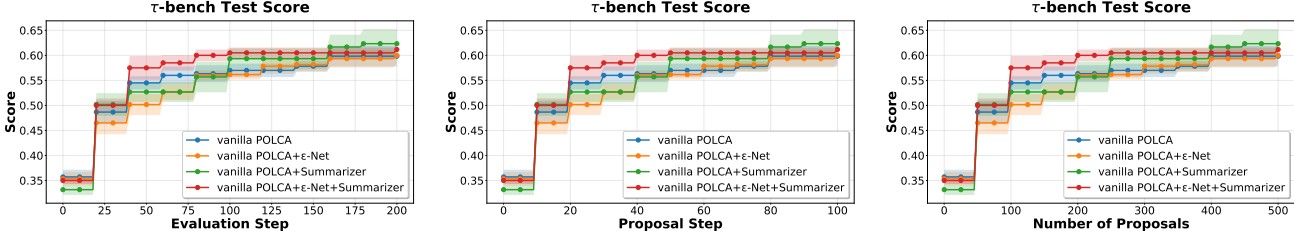

*Figure 12.* Ablation on $\varepsilon$-Net and Summarizer. Solid curves represent the average highest score attained at each step, while the shaded regions denote the standard error across multiple independent runs (6 seeds).

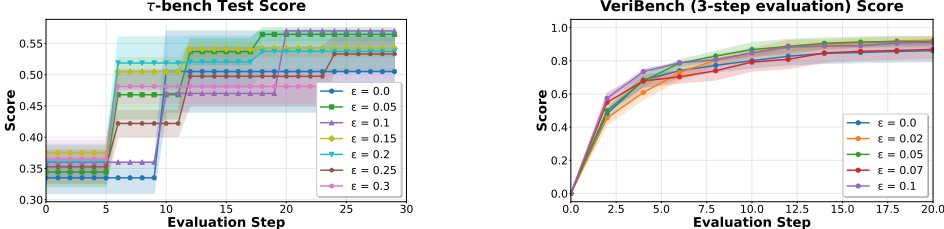

*Figure 13.* Ablation study on $\varepsilon$-values ($\tau$-bench and VeriBench). Solid curves represent the average highest score attained at each step, while shaded regions denote the standard error across 3 independent seeds.

**Additional experiments with open-source model evaluation.** To further demonstrate the generality of POLCA across model families, we run additional experiments with weaker open-source backbones: HotpotQA with `gpt-oss-20b` (Agarwal et al., 2025) and VeriBench (Compilation) with `Qwen3.5-122B-A10B` (Team, 2026). The results are shown in Figures 14 and 15: even with these weaker backbones, POLCA still consistently outperforms the baselines across all four budget axes (evaluation steps, metric calls, proposal steps, and number of proposals).

Combined with our main experiments, we have now evaluated POLCA on `gemini-2.0-flash`, `gemini-2.5-flash-lite`, `claude-3.5-sonnet`, `claude-3.7-sonnet`, `gpt-oss-20b`, and `Qwen3.5-122B-A10B` across $\tau$-bench, HotpotQA, VeriBench, and KernelBench, covering a broad range of model families and application domains.

### E.8. Why Not Use Regression?

In large-scale generative optimization, the search process often yields reward observations for a wide variety of programs. A naive approach is to evaluate each program separately and estimate its reward independently. However, due to the inherent stochasticity of these observations, such an approach can require a prohibitively large evaluation budget to achieve precision.

**Embedding similarity is locally informative.** We first observe that the program embedding space is locally smooth: candidates with similar embeddings tend to have similar scores. Figure 16 shows a t-SNE projection of prompt embeddings collected during a HotpotQA run, with each point colored by its test score—tightly clustered points share similar colors,

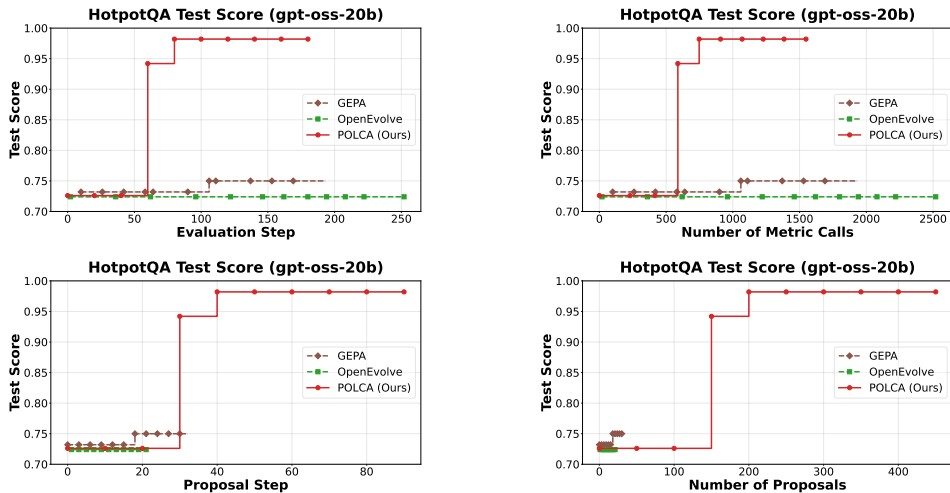

*Figure 14.* HotpotQA test scores with the `gpt-oss-20b` backbone, plotted against evaluation steps, metric calls, proposal steps, and number of proposals. Even with this weaker open-source model, POLCA consistently outperforms the baselines.

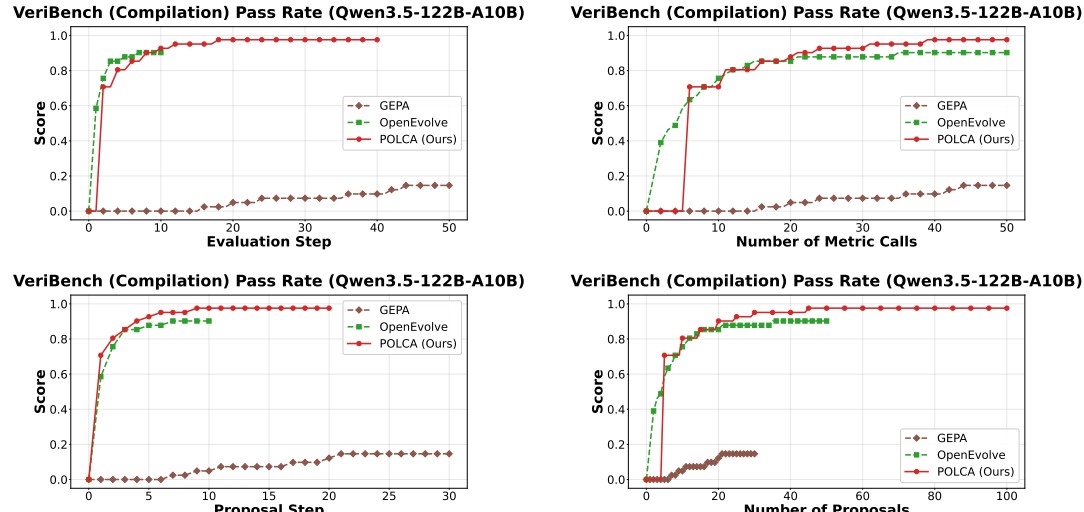

*Figure 15.* VeriBench (Compilation) compilation rates with the `Qwen3.5-122B-A10B` backbone, plotted against evaluation steps, metric calls, proposal steps, and number of proposals. POLCA remains the strongest method across all budget axes.

indicating that local embedding distance is a meaningful proxy for reward similarity. This justifies the $\varepsilon$-Net filtering mechanism in POLCA, which uses embedding similarity to discard near-duplicate candidates during optimization.

Given this local smoothness, it is natural to ask whether a stronger *global* use of the embedding space is possible: can we train a regression model on (embedding, reward) pairs and use its predictions to rank candidates, in place of empirical means? In the rest of this section we show that, despite the local smoothness above, such a global regressor does *not* reliably generalize across the program space, and the empirical mean remains the more robust priority. Concretely, we use function approximation by training a reward model that maps a program's embedding to its expected reward:

$$\psi \circ \phi : \Theta \to \mathbb{R},$$

where $\phi$ maps a program to an embedding vector, and $\psi$ predicts a reward in $[0, 1]$. The goal is for such a model to generalize across the program space, leveraging the entire dataset to estimate the reward of any given program. This is particularly appealing because similar programs likely yield similar rewards; thus, a learned predictor might provide more robust estimates than a noisy empirical mean, especially for programs with few observations.

## Prompt Embedding Space

*Figure 16.* t-SNE visualization of prompt embeddings collected during a HotpotQA optimization run. Each point is a candidate program; color encodes its test score. Programs with similar embeddings (forming visual clusters) tend to share similar scores, supporting the use of embedding similarity in our $\varepsilon$-Net filter.

To train this model, we collect a dataset of embedding–reward pairs

$$\{(\phi(\theta_j), r_j)\}_{j=1}^N,$$

where each reward observation $r_j \in \{0, 1\}$ is binary. We instantiate $\psi$ as a logistic regression model parameterized by $(w, b)$:

$$\psi(z) = \sigma(w^\top z + b), \qquad z \in \mathbb{R}^d,$$

where $\sigma(z) = 1/(1 + e^{-z})$ is the sigmoid function. The predicted reward for a program $\theta$ is then:

$$(\psi \circ \phi)(\theta) = \sigma(w^\top \phi(\theta) + b).$$

We optimize the parameters $(w, b)$ by minimizing the regularized empirical risk:

$$\mathcal{L}(w, b) = -\frac{1}{N} \sum_{j=1}^N \left[ r_j \log \sigma(w^\top \phi(\theta_j) + b) + (1 - r_j) \log\big(1 - \sigma(w^\top \phi(\theta_j) + b)\big) \right] + \frac{\lambda}{2} \|w\|_2^2,$$

where $\lambda > 0$ is the regularization parameter.

To further reduce variance and enhance diversity, we employ an ensemble of five regressors, each trained on a randomly sampled subset of the data. These models predict scores for all programs stored in the priority memory $\mathcal{Q}$. For each program $\theta \in \mathcal{Q}$, let $\widehat{p}_k(\theta)$ be the prediction of the $k$-th regressor and $\widehat{\mu}(\theta)$ be the empirical mean reward based on its existing samples. We then compute four candidate scores:

$$S_{\text{emp}}(\theta) = \widehat{\mu}(\theta),$$
$$S_{\max}(\theta) = \max_{1 \le k \le 5} \widehat{p}_k(\theta),$$
$$S_{\text{mean}}(\theta) = \frac{1}{5} \sum_{k=1}^5 \widehat{p}_k(\theta),$$
$$S_{\min}(\theta) = \min_{1 \le k \le 5} \widehat{p}_k(\theta).$$

Across various training set sizes $N$, we select the program with the highest score for each metric and subject it to additional external testing for a more precise reward estimate. However, our results (Figure 4) show that the naive strategy—selecting programs based solely on the empirical mean—consistently outperforms all regressor-based criteria. This suggests that, in this specific optimization setting, the regression model fails to provide sufficiently reliable generalization to surpass simple empirical estimation.

### E.9. Token Usage Estimation

We evaluate the token consumption of the generative optimization process by monitoring a single run of POLCA, using the configuration detailed in Section E.2, across the $\tau$-bench, HotpotQA, VeriBench, and KernelBench benchmarks. This estimation is limited to the tokens utilized by the search pipeline and excludes LLM calls invoked within the optimizing programs themselves.

For $\tau$-bench, a single run (100 iterations) requires 30,822,470 input tokens and 380,240 output tokens, totaling 31,202,710 tokens. For HotpotQA, a single run (100 iterations) requires 4,739,242 input tokens and 1,801,051 output tokens, totaling 6,540,293 tokens. For VeriBench, a single run (10 iterations) on a single task requires 554,900 input tokens and 79,145 output tokens, totaling 634,045 tokens. For KernelBench, a single run (10 iterations) on a single task requires 477,186 input tokens and 93,432 output tokens, totaling 570,618 tokens.

## F. Optimized Program Case Studies

In this section, we provide representative examples of programs optimized using POLCA on the $\tau$-bench, VeriBench, and KernelBench benchmarks. These cases illustrate the ability of our generative optimization framework to discover high-performing solutions across diverse domains, from multi-turn agentic reasoning to formal verification and hardware-level performance tuning.

### F.1. $\tau$-bench

In $\tau$-bench, we formalize the search problem by appending a trainable string to the original system prompt of the agent as *parameters*; this string is initialized as the *original instruction prompt* and trained using generative optimization algorithms.

---

**Original Instruction Prompt**

Here are the additional instructions to help the agent solve the task:

---

**Trained Prompt from POLCA**

Here are the additional instructions to help the agent solve the task:

- **IMPORTANT: Only ONE order can be modified or exchanged per conversation. Inform the user of this limit at the beginning of the interaction.** If the user mentions modifying items in multiple orders, inform them that only ONE order can be modified. Ask them to choose the order they want to modify. Then proceed with modifying only that order. Remind the user of this limitation before presenting any product options or proceeding with any modifications. **Begin the conversation by stating this limitation.**

- Prioritize verifying user identity at the beginning of the conversation, with email as the primary method. If email verification fails, *double-check the email address provided with the user*, and if it's incorrect, offer alternative authentication methods (first name, last name, zip code) *only after email verification definitively fails*. Implement a more robust retry mechanism if the initial attempt fails, retrying up to three times. *If email verification persistently fails after multiple attempts, proceed with name/zip code verification but inform the user that email verification is preferred for security purposes.* Continue verifying identity throughout the conversation. Do not proceed with any order-related actions until the user's identity is confirmed. *If the user is successfully identified by name and zip code, offer to update their user profile with the correct email address, if available, to streamline future interactions. Explicitly offer to save the email address to the user profile to streamline future interactions. If the user provides a new email address, use* update_user_email *tool to update the user's profile. ALWAYS update the user profile with the email address if they are authenticated via name/zip.* Only attempt user identification by name/zip after email verification has failed. *Explicitly state the importance of providing accurate information for successful identification.* If the user has privacy concerns, explain that identity verification is for their protection and to prevent unauthorized access to their account. If the email is incorrect or not found, *provide a specific error message to the user, such as: "I'm sorry, but I couldn't find a user with that email address. Could you please double-check the email address you provided?*

*It's important to provide accurate information for successful identification. Please confirm your email or provide your first name, last name and zip code."*

- **Immediately after user identification, before proceeding with any requests, offer the user a summary of available actions:** "Okay, Yusuf, now that I've verified your identity, I can help you with the following: modify a pending order, cancel a pending order, return a delivered order, or exchange a delivered order. Which action would you like to take?"

- When the user asks to modify or exchange an order, especially regarding item characteristics (color, size, material, style, brightness, compatibility), *proactively* ask for the order ID *and the item IDs of the items they want to modify or exchange*. Remind the user that only a single exchange or modification call is available per order, so it's important to include ALL desired changes in one request. Gather ALL the necessary details about the desired options BEFORE attempting to find matching items. Remind the user that exchange or modify order tools can only be called once per order. *Specifically, confirm the item ID, the attributes of the old items, and the attributes of the new items.* If the user is unsure which order contains the items, offer to search their order history using their user ID and present the contents of each order to the user to help them identify the correct one.

- Immediately after the user provides an order ID, call `get_order_details` to proactively validate the order's existence, contents, and status. *Flag any discrepancies* between the user's request and the actual order details and ask them to verify. Clarify any discrepancies with the user before proceeding. *Specifically, confirm that ALL items the user wants to modify, exchange, or return are actually in the identified order.* **Before confirming the items, ALWAYS check the order status to ensure the requested action (modification, exchange, return) is possible given the current order status.** *Present the order details (contents, status, shipping address) to the user and ask them to confirm everything is accurate before proceeding. Check if the order status even supports the request (e.g. pending for modifications, delivered for exchanges or returns).* If the order has already been delivered or cancelled, inform the user that you cannot modify it. If the user doesn't know the order ID or item IDs, *proactively* offer to retrieve order details using their user ID. *If the user provides an order ID but the items they mention are not in that order, ask the user to double-check the order ID and confirm if the listed items are indeed in that order. Validate the order details and explicitly ask the user to confirm the correctness of item names, quantities, and attributes.* **Before listing product types for returns or exchanges, confirm the order ID and item IDs related to the return or exchange.**

- If a tool, such as `search_items`, fails or returns an error, inform the user that the tool is unavailable and offer an alternative approach. *DO NOT use `search_items` as it is currently unreliable. Prioritize using `get_product_details` to obtain specific item details before resorting to `list_all_product_types`.* If the `search_items` tool fails, you can try offering an alternative solution, such as connecting the user with a human agent or using different search parameters. However, transfer only after exhausting all available strategies and tool calls. If a tool fails, inform the user that the tool is unavailable and offer an alternative approach. For example, offer to list available options for each attribute (color, size, material, style) individually using `get_product_details` and ask the user to choose from the available options. If `get_product_details` also fails, suggest that the user browse our online catalog and provide the item ID. Do NOT guess or proceed without confirming the correct item ID. *If `list_all_product_types` returns an empty dictionary, inform the user that there are currently no products of that type available and apologize for the inconvenience.*

- When presenting product options for an exchange or modification, always use the information from `get_product_details` and include the item ID, features, price, and availability. If an item is unavailable, clearly state that and offer alternatives. Do not mention that the product options are unavailable until all product options are displayed. When presenting options, display a maximum of 5 options at a time for better user experience.

- Always clarify with the user BEFORE using `exchange_items` or `modify_pending_order_items`. *Explicitly state and confirm all identified items and their attributes, and the desired changes, before using the tool.* Before calling the tool, repeat the action being taken: "You are exchanging item A for item B, and your card C will be charged $X. Is this correct?". Always confirm the exchange/modification details with the user before executing

the final tool call. This confirmation should include the items being exchanged/modified, the new items (with all their characteristics), and the payment method for any price difference. *Remind the user that exchange and modify order tools can only be called once, so all desired changes must be included in a single request. After the tool call, repeat the confirmation with the user before finalizing. Ensure the agent gathers all necessary information before attempting to call the* `exchange_items` *or* `modify_pending_order_items` *tool.*

- Be sure to check the order status before taking actions like cancelling, modifying, returning, or exchanging. Actions are generally only possible on pending or delivered orders. *Clarify which actions are available for each order status.* Implement robust order verification steps to ensure the identified order matches the user's intended items. *Specifically, for returns, ensure the order status is 'delivered' before attempting to process the return.*

- Before asking for a payment method, process all product options and confirm the user's selection. *A payment method is only required if there is a price difference or if required by the tool itself. Do not ask for a payment method if there is no price difference and the tool doesn't explicitly require it.*

- *Always validate with the user: Validate all details of the identified order before making changes, clarify any ambiguities before acting.*

- When dealing with returns, *immediately ask for the order ID and item IDs.* Then, confirm all of the information, like order id and items to be returned, before listing all of the product types.

- Handle multiple requests or orders within a single conversation step-by-step. Complete one action fully before moving on to the next. If the user is indecisive, summarize the current state, options, and consequences to guide them. *If the user changes scope often, remind the user of the single-exchange/modification limit to guide them. If the user is indecisive or changes their mind frequently, summarize the confirmed details (order ID, items, requested actions) and require explicit confirmation before proceeding.*

- If the user provides an incorrect order ID, politely inform them of the potential mistake and offer to look up their orders using their user ID to confirm the correct order ID. If the user provides the wrong item IDs, offer the user to find the item IDs on the website and suggest finding them now before moving on.

- *Implement a loop to help the user select and identify the correct item by cross-checking the details provided by the user or other items in the order.*

- *When the user asks for the number of available options for a product (e.g., t-shirts), use* `get_product_details` *to retrieve the product's variants. Count the number of variants where "available": true, and then inform the user of the total count of available options.* Prioritize answering this request before handling other requests from the user. **Count the number of variants where "available": true, and then inform the user of the total count of available options. In the example, there are 10 t-shirt options currently available. There are 12 variants listed, and 2 of them are unavailable.**

*After collecting item and order details, create a summary of all confirmed items, attributes, desired changes, and the payment method. Require explicit user confirmation (e.g., "So, to confirm, you want to exchange item A (size M, blue) for item B (size L, red), and the price difference of $X will be charged to your card ending in 1234. Is that correct?") before proceeding.*

- **If the user mentions modifying items in multiple orders, inform them that only ONE order can be modified. Ask them to choose the order they want to modify. Then proceed with modifying only that order.**

- **Before proceeding with modifications, ALWAYS check the order status. If the order status is not 'pending', inform the user that the order cannot be modified.**

*Before calling any tool, the agent should say, "Just to confirm, I have the following information: ... Is this correct?"* If any information is missing, ask the user to provide it before proceeding.

### F.2. HotpotQA

In HotpotQA, we optimize the prompt for the question-answering task. Below, we present the *original prompt* alongside a *trained prompt* generated by POLCA.

---

**Original Prompt**

Answer the question based on the context.

---

**Trained Prompt from POLCA**

Carefully read the provided context and the question. Systematically break down the question into its constituent parts and identify all required information. **For multi-hop reasoning questions, meticulously document the entire reasoning path before providing the final answer.** This path must:

1. Deconstruct the question into a series of sub-questions or identify key intermediate facts/entities required.

2. Detail the retrieval of relevant information for each sub-question or intermediate component.

3. **Explicitly demonstrate the logical connections and dependencies between the retrieved pieces of information, forming a coherent reasoning chain.** For example, explain how Fact A leads to Fact B, which is then used with Fact C to infer a conclusion.

4. Synthesize these interconnected facts and intermediate conclusions to directly address and answer the main question.

**Answer Formatting Rules:**

1. **Direct Retrieval Questions:** If a question solely asks for a specific entity name, numerical value, or factual item, provide **only** that exact, complete identifier or fact as presented in the context. Do not include any preamble, explanation, or supporting details.

2. **Entity Identification:** For questions asking to identify an entity, the primary answer must be the exact, complete identifier as presented in the context. This primary answer must be stated first and without preamble. If the question implies a need for elaboration beyond the direct identification, supporting details from the context should follow.

3. **Concept/Practice/Method Identification:** For questions asking to identify a practice, method, or concept, the primary answer must be the most appropriate conceptual term or category from the context, stated first and without preamble. Supporting details from the context that elaborate on this concept should follow. Avoid answering with only descriptive details if a specific term exists.

4. **Comparative Questions (Locations):** Explicitly analyze and differentiate between geographical levels (e.g., city, district, neighborhood). Do not infer 'same' based solely on a broader shared context.

5. **Numerical Data:**
   - Prioritize specific, qualified figures.
   - If the question directly asks for a numerical value, present the most specific, qualified figure available in the context upfront as the primary answer, without preamble.
   - If the requested scope (e.g., country population) is not directly available but a relevant sub-unit's data is (e.g., county population), first state that the requested broad-scope information is not available. Then, provide the qualified sub-unit figure, clearly qualifying it as belonging to that sub-unit.
   - Clearly qualify all numerical data presented.

6. **Binary Answers (Yes/No):** Provide the direct binary answer clearly and upfront, followed by supporting details that justify the conclusion.

---

7. **Handling Questions with Flawed Premises:** If the question's premise contains a factual inaccuracy (e.g., refers to a song as an album), the agent must address this directly. The response should:

   - State that the requested entity type (e.g., "album") is not found as described in the context.
   - Identify the correct entity type (e.g., "song") and provide its relevant details from the context.
   - If applicable, provide information about the containing album (name and release date), clearly distinguishing it from the song.
   - This structured explanation takes precedence over Rule 1 when a flawed premise is detected.

8. **Handling Ambiguity and Underspecified Questions:**

   - If a question can have multiple valid answers based on the context, first attempt to resolve the ambiguity using the following prioritized strategies:
     1. **Prioritize Explicit/Unique Links:** Select the option most directly or uniquely linked to the question's criteria.
     2. **Prioritize Prominence/Emphasis:** If the context highlights a specific entity, use this as a tie-breaker.
     3. **Prioritize Order of Appearance:** If no other criteria apply, select the first valid option encountered in the text.
   - If a definitive single choice cannot be made after applying these strategies, state that multiple answers exist and list them clearly, qualifying that the question was underspecified. For example: "There are multiple valid answers fitting this description. One is [Answer A]. Another is [Answer B]."

Provide a complete, fully qualified, and verifiable answer that directly addresses all parts of the question, synthesized from the context. Ensure answers are comprehensive and include all necessary qualifiers, especially for location-based queries.

## F.3. VeriBench

In VeriBench, the goal is to translate Python programs into verifiable Lean 4 programs. We utilize generative optimization algorithms to optimize for these Lean 4 programs directly, iteratively refining the formal code to satisfy verification requirements. Below, we provide a concrete example (square function) of this process.

**Input Python Program**

```python
"""Return the square of a natural number (non-negative integer).

Edge cases:
- square(0) = 0
- square(1) = 1
- The function raises ValueError if input is not a natural number.
"""

def pre(n: int) -> bool:
    """True iff the input is a non-negative integer."""
    return isinstance(n, int) and n >= 0

def prog(n: int) -> int:
    """
    Return n² for non-negative integer n.
    Raises ValueError if the precondition is violated.
    """
    if not pre(n):
        raise ValueError(f"Input must be a non-negative integer (got n={n})")
    return n * n

# -- Tests --
```

```
from typing import Callable

def check(candidate: Callable[[int], int]) -> bool:
    # Basic unit test
    assert candidate(3) == 9

    # Edge unit test
    assert candidate(0) == 0

    # Negative (pre-violations must raise ValueError)
    bad_inputs = [-1, 3.14]
    for n in bad_inputs:
        try:
            candidate(n)
            raise AssertionError("expected pre-violation did not raise")
        except ValueError:
            pass

    return True

if __name__ == "__main__":
    assert check(prog), f"Failed: {__file__}"
    print("All tests passed.")
```

**Initial Lean 4 Program**

```
    -- Lean 4 translation of the Python program
```

**Trained Lean 4 Program**

```
/-!
# Square Function

This module implements a square function for natural numbers.

## Features:
- Returns square of non-negative integers
- Edge cases: square(0) = 0, square(1) = 1
- Input validation for natural numbers
-/

namespace Square

/--
**Implementation of `square`.**

`square n` returns the natural number n².

## Examples
#eval square 3 -- expected: 9
#eval square 0 -- expected: 0
-/
def square (n : Nat) : Nat := n * n

/-!
# Tests
-/

/-- Basic test: 3^2 = 9 -/
example : square 3 = 9 := by native_decide
#eval square 3 -- expected: 9
```

```
/-!
# Tests: Edge Cases
-/

/-- Edge case: 0^2 = 0 -/
example : square 0 = 0 := by native_decide
#eval square 0 -- expected: 0

/-- Edge case: 1^2 = 1 -/
example : square 1 = 1 := by native_decide
#eval square 1 -- expected: 1

/-!
# Positive / Negative Test-Suite
-/

/-- positive: 4^2 = 16 -/
example : square 4 = 16 := by native_decide
#eval square 4 -- expected: 16

/-- positive: 5^2 = 25 -/
example : square 5 = 25 := by native_decide
#eval square 5 -- expected: 25

/-- negative: 2^2 != 5 -/
example : ¬(square 2 = 5) := by native_decide
#eval (decide (square 2 = 5)) -- expected: false

/-- negative: 3^2 != 8 -/
example : ¬(square 3 = 8) := by native_decide
#eval (decide (square 3 = 8)) -- expected: false

/-!
# Pre-Condition
-/

/-- **Pre-condition.** Input is non-negative (always true for Nat). -/
def Pre (n : Nat) : Prop := 0 <= n

/-!
# Property Theorems
-/

/-- **Non-negativity property**: square of any natural number is non-negative. -/
def non_negative_prop (n : Nat) : Prop := 0 <= square n

/-- **Non-negativity theorem** -/
theorem non_negative_thm (n : Nat) : non_negative_prop n := sorry

/-- **Zero property**: square of zero is zero. -/
def zero_prop : Prop := square 0 = 0

/-- **Zero theorem** -/
theorem zero_thm : zero_prop := sorry

/-- **One property**: square of one is one. -/
def one_prop : Prop := square 1 = 1

/-- **One theorem** -/
theorem one_thm : one_prop := sorry

/-- **Multiplication property**: square equals number multiplied by itself. -/
def mult_prop (n : Nat) : Prop := square n = n * n
```

```
/-- **Multiplication theorem** -/
theorem mult_thm (n : Nat) : mult_prop n := sorry

/-!
# Post-Condition
-/

/-- **Post-condition**: conjunction of all desired properties for square. -/
def Post_prop (n : Nat) : Prop :=
  (non_negative_prop n) /\
  (mult_prop n) /\
  zero_prop /\
  one_prop

/-!
# Correctness Theorem
-/

/-- **Correctness theorem**: the pre-condition implies the post-condition. -/
theorem correctness_thm (n : Nat) (hPre : Pre n) : Post_prop n := sorry

/-!
# Imperative Implementation
-/

/--
`squareImp n` computes the same square using accumulation.
-/
def squareImp (n : Nat) : Nat :=
Id.run do
  let mut acc : Nat := 0
  for _ in [:n] do
    acc := acc + n
  return acc

/-!
# Imperative Tests
-/

/-- Basic test: imp 3^2 = 9 -/
example : squareImp 3 = 9 := by native_decide
#eval squareImp 3 -- expected: 9

/-- Edge case: imp 0^2 = 0 -/
example : squareImp 0 = 0 := by native_decide
#eval squareImp 0 -- expected: 0

/-- Edge case: imp 1^2 = 1 -/
example : squareImp 1 = 1 := by native_decide
#eval squareImp 1 -- expected: 1

/-!
# Implementation Equivalence
-/

/-- **Equivalence theorem**: functional and imperative implementations
coincide. -/
theorem square_equivalence_thm (n : Nat) :
square n = squareImp n := sorry

end Square
```

## F.4. KernelBench

In KernelBench, the goal is to optimize custom CUDA kernels to outperform baseline PyTorch implementations in terms of execution speed. Here, we provide a concrete example (batched matrix multiplication) of this optimization process.

---

**Task Prompt**

```
You write custom CUDA kernels to replace the pytorch operators in the given
architecture to get speedups.

You have complete freedom to choose the set of operators you want to replace.
You may make the decision to replace some operators with custom CUDA kernels
and leave others unchanged.
You may replace multiple operators with custom implementations,
consider operator fusion opportunities (combining multiple operators into a
single kernel, for example, combining matmul+relu), or
algorithmic changes (such as online softmax).
You are only limited by your imagination.

Here's an example to show you the syntax of inline embedding custom CUDA
operators in torch: The example given architecture is:

```

import torch
import torch.nn as nn
import torch.nn.functional as F

class Model(nn.Module):
    def __init__(self) -> None:
        super().__init__()

    def forward(self, a, b):
        return a + b

def get_inputs():
    # randomly generate input tensors based on the model architecture
    a = torch.randn(1, 128).cuda()
    b = torch.randn(1, 128).cuda()
    return [a, b]

def get_init_inputs():
    # randomly generate tensors required for initialization based on the model
    architecture
    return []

```

The example new arch with custom CUDA kernels looks like this:
```
import torch
import torch.nn as nn
import torch.nn.functional as F
from torch.utils.cpp_extension import load_inline

# Define the custom CUDA kernel for element-wise addition
elementwise_add_source = """
#include <torch/extension.h>
#include <cuda_runtime.h>
```

```
__global__ void elementwise_add_kernel(const float* a, const
float* b, float* out, int size) {
    int idx = blockIdx.x * blockDim.x + threadIdx.x;
    if (idx < size) {
        out[idx] = a[idx] + b[idx];
    }
}

torch::Tensor elementwise_add_cuda(torch::Tensor a, torch::Tensor b) {
    auto size = a.numel();
    auto out = torch::zeros_like(a);

    const int block_size = 256;
    const int num_blocks = (size + block_size - 1) / block_size;

    elementwise_add_kernel<<<num_blocks, block_size>>>
    (a.data_ptr<float>(), b.data_ptr<float>(), out.data_ptr<float>(), size);

    return out;
}
"""

elementwise_add_cpp_source = (
    "torch::Tensor elementwise_add_cuda(torch::Tensor a, torch::Tensor b);"
)

# Compile the inline CUDA code for element-wise addition
elementwise_add = load_inline(
    name="elementwise_add",
    cpp_sources=elementwise_add_cpp_source,
    cuda_sources=elementwise_add_source,
    functions=["elementwise_add_cuda"],
    verbose=True,
    extra_cflags=[""],
    extra_ldflags=[""],
)

class ModelNew(nn.Module):
    def __init__(self) -> None:
        super().__init__()
        self.elementwise_add = elementwise_add

    def forward(self, a, b):
        return self.elementwise_add.elementwise_add_cuda(a, b)

```

You are given the following architecture:

```
import torch
import torch.nn as nn

class Model(nn.Module):
    """
    Performs batched matrix multiplication (C = A * B)
    where A, B, and C have the same batch dimension.
    """
    def __init__(self):
        super(Model, self).__init__()
```

```
    def forward(self, A: torch.Tensor, B: torch.Tensor) -> torch.Tensor:
        """
        Performs batched matrix multiplication.

        Args:
            A: Input tensor of shape (batch_size, m, k).
            B: Input tensor of shape (batch_size, k, n).

        Returns:
            C: Output tensor of shape (batch_size, m, n).
        """
        return torch.bmm(A, B)

batch_size = 128
m = 128 * 4
k = 256 * 4
n = 512 * 4

def get_inputs():
    A = torch.rand(batch_size, m, k)
    B = torch.rand(batch_size, k, n)
    return [A, B]

def get_init_inputs():
    return []  # No special initialization inputs needed
```

Optimize the architecture named Model with custom CUDA operators!
Name your optimized output architecture ModelNew. Output the new code in
codeblocks. Please generate real code, NOT pseudocode, make sure the code
compiles and is fully
functional. Just output the new model code, no other text, and NO testing code!

## Initial CUDA Kernel

```
# This is a dummy kernel code. You should replace it with your own kernel code
based on the task prompt and optimization objectives.
```

## Trained CUDA Kernel

```python
import torch
import torch.nn as nn
from torch.utils.cpp_extension import load_inline

# Define the custom CUDA kernel for batched matrix multiplication
batched_matmul_source = """
#include <torch/extension.h>
#include <cuda_runtime.h>
#include <cublas_v2.h>

#define CHECK_CUDA(x) TORCH_CHECK(x.is_cuda(), #x " must be a CUDA tensor")
#define CHECK_CONTIGUOUS(x) TORCH_CHECK(x.is_contiguous(), #x " must be contiguous")
#define CHECK_CUBLAS(err) { cublasStatus_t status = (err); if
(status != CUBLAS_STATUS_SUCCESS) { throw std::runtime_error("cuBLAS error: " +
std::to_string(status)); } }

// Singleton class to manage cuBLAS handle with TENSOR_OP math mode for automatic
Tensor Core usage
class CuBLASHandleManager {
public:
```

```cpp
    static cublasHandle_t& getHandle() {
        static CuBLASHandleManager instance;
        return instance.handle;
    }

private:
    cublasHandle_t handle;

    CuBLASHandleManager() {
        CHECK_CUBLAS(cublasCreate(&handle));
        // Enable Tensor Cores if available
        CHECK_CUBLAS(cublasSetMathMode(handle, CUBLAS_TENSOR_OP_MATH));
        // Set stream for better asynchronous execution
        cudaStream_t stream;
        cudaStreamCreateWithFlags(&stream, cudaStreamNonBlocking);
        CHECK_CUBLAS(cublasSetStream(handle, stream));
    }

    ~CuBLASHandleManager() {
        cudaStream_t stream;
        cublasGetStream(handle, &stream);
        cublasDestroy(handle);
        if (stream) cudaStreamDestroy(stream);
    }
};

// Optimized batched matrix multiplication with specialized handling for common shapes
torch::Tensor batched_matmul_cuda(torch::Tensor A, torch::Tensor B) {
    // Quick checks before proceeding
    CHECK_CUDA(A);
    CHECK_CUDA(B);

    const int batch_size = A.size(0);
    const int m = A.size(1);
    const int k = A.size(2);
    const int n = B.size(2);

    // Create output tensor
    auto C = torch::empty({batch_size, m, n}, A.options());

    // Ensure input tensors are contiguous for best performance
    auto A_cont = A.contiguous();
    auto B_cont = B.contiguous();

    // Get data pointers
    float* A_data = A_cont.data_ptr<float>();
    float* B_data = B_cont.data_ptr<float>();
    float* C_data = C.data_ptr<float>();

    // Get cuBLAS handle
    cublasHandle_t handle = CuBLASHandleManager::getHandle();

    // Constants for cuBLAS call
    const float alpha = 1.0f;
    const float beta = 0.0f;

    // Use strided batch GEMM for maximum performance
    // cuBLAS uses column-major order while PyTorch uses row-major
    // To compute C = A * B in row-major, we compute C^T = B^T * A^T in column-major

    // Use cublasGemmStridedBatchedEx for better performance (supports mixed precision)
    CHECK_CUBLAS(cublasGemmStridedBatchedEx(
        handle,
```

```
            CUBLAS_OP_N, CUBLAS_OP_N,
            n, m, k,                            // Dimensions for column-major (swapped)
            &alpha,
            B_data, CUDA_R_32F, n, k * n,  // B, data type, leading dimension, stride
            A_data, CUDA_R_32F, k, m * k,  // A, data type, leading dimension, stride
            &beta,
            C_data, CUDA_R_32F, n, m * n,  // C, data type, leading dimension, stride
            batch_size,                         // Number of matrices
            CUDA_R_32F,                         // Computation type
            CUBLAS_GEMM_DEFAULT_TENSOR_OP // Algorithm selection
    ));

    return C;
}

// Half-precision version for tensor cores
torch::Tensor batched_matmul_half_cuda(torch::Tensor A, torch::Tensor B) {
    CHECK_CUDA(A);
    CHECK_CUDA(B);

    const int batch_size = A.size(0);
    const int m = A.size(1);
    const int k = A.size(2);
    const int n = B.size(2);

    // Convert to half precision if needed
    auto A_half = A.type().scalarType() == torch::ScalarType::Half ? A.contiguous() :
    A.contiguous().to(torch::ScalarType::Half);
    auto B_half = B.type().scalarType() == torch::ScalarType::Half ? B.contiguous() :
    B.contiguous().to(torch::ScalarType::Half);

    // Create output tensor in half precision
    auto C_half = torch::empty({batch_size, m, n}, A_half.options());

    // Get data pointers
    __half* A_data = reinterpret_cast<__half*>(A_half.data_ptr<torch::Half>());
    __half* B_data = reinterpret_cast<__half*>(B_half.data_ptr<torch::Half>());
    __half* C_data = reinterpret_cast<__half*>(C_half.data_ptr<torch::Half>());

    // Get cuBLAS handle
    cublasHandle_t handle = CuBLASHandleManager::getHandle();

    // Constants for cuBLAS call
    const __half alpha = __float2half(1.0f);
    const __half beta = __float2half(0.0f);

    // Use cublasGemmStridedBatchedEx for better performance and more flexibility with
    half precision
    CHECK_CUBLAS(cublasGemmStridedBatchedEx(
        handle,
        CUBLAS_OP_N, CUBLAS_OP_N,
        n, m, k,
        &alpha,
        B_data, CUDA_R_16F, n, k * n,
        A_data, CUDA_R_16F, k, m * k,
        &beta,
        C_data, CUDA_R_16F, n, m * n,
        batch_size,
        CUDA_R_16F,
        CUBLAS_GEMM_DEFAULT_TENSOR_OP
    ));
```

```cpp
    // Convert back to original precision if needed
    return A.type().scalarType() == torch::ScalarType::Half ? C_half :
    C_half.to(A.type().scalarType());
}

// INT8 quantized version for maximum throughput on compatible hardware
torch::Tensor batched_matmul_int8_cuda(torch::Tensor A, torch::Tensor B) {
    CHECK_CUDA(A);
    CHECK_CUDA(B);

    // Original shapes
    const int batch_size = A.size(0);
    const int m = A.size(1);
    const int k = A.size(2);
    const int n = B.size(2);

    // Quantize to INT8 (this is a simplified quantization)
    auto A_float = A.contiguous();
    auto B_float = B.contiguous();

    // Scale factors for quantization/dequantization
    float scale_A = 127.0f / A_float.abs().max().item<float>();
    float scale_B = 127.0f / B_float.abs().max().item<float>();
    float scale_C = 1.0f / (scale_A * scale_B);

    // Create quantized tensors
    auto A_int8 = (A_float * scale_A).to(torch::kChar).contiguous();
    auto B_int8 = (B_float * scale_B).to(torch::kChar).contiguous();

    // Output will be in int32
    auto C_int32 = torch::empty({batch_size, m, n},
            torch::TensorOptions().dtype(torch::kInt).device(A.device()));

    // Get data pointers
    int8_t* A_data = A_int8.data_ptr<int8_t>();
    int8_t* B_data = B_int8.data_ptr<int8_t>();
    int* C_data = C_int32.data_ptr<int>();

    // Get cuBLAS handle
    cublasHandle_t handle = CuBLASHandleManager::getHandle();

    // Constants for INT8 GEMM
    const int32_t alpha = 1;
    const int32_t beta = 0;

    // Use INT8 GEMM with tensor cores if available
    // Note: This is a simplified implementation; real-world would need more careful
    quantization
    CHECK_CUBLAS(cublasGemmStridedBatchedEx(
        handle,
        CUBLAS_OP_N, CUBLAS_OP_N,
        n, m, k,
        &alpha,
        B_data, CUDA_R_8I, n, k * n,
        A_data, CUDA_R_8I, k, m * k,
        &beta,
        C_data, CUDA_R_32I, n, m * n,
        batch_size,
        CUDA_R_32I,
        CUBLAS_GEMM_DEFAULT_TENSOR_OP
    ));

    // Dequantize and return
```

```
    auto C_float = C_int32.to(torch::kFloat) * scale_C;
    return C_float;
}
"""

batched_matmul_cpp_source = """
torch::Tensor batched_matmul_cuda(torch::Tensor A, torch::Tensor B);
torch::Tensor batched_matmul_half_cuda(torch::Tensor A, torch::Tensor B);
torch::Tensor batched_matmul_int8_cuda(torch::Tensor A, torch::Tensor B);
"""

# Compile the inline CUDA code for batched matrix multiplication
custom_ops = load_inline(
    name="batched_matmul_ops",
    cpp_sources=batched_matmul_cpp_source,
    cuda_sources=batched_matmul_source,
    functions=[
        "batched_matmul_cuda",
        "batched_matmul_half_cuda",
        "batched_matmul_int8_cuda",
    ],
    verbose=True,
    extra_ldflags=["-lcublas"],
    extra_cuda_cflags=[
        "--expt-relaxed-constexpr",
        "-O3",
        "--use_fast_math",
        "-Xptxas=-v",
    ],
)

class ModelNew(nn.Module):
    """
    Optimized version of the model that performs batched matrix multiplication
    using custom CUDA kernels with cuBLAS. The implementation includes options
    for half-precision and INT8 computation to leverage tensor cores when available.
    """

    def __init__(self):
        super(ModelNew, self).__init__()
        self.custom_ops = custom_ops
        self.use_half_precision = False  # Can be toggled based on hardware support
        self.use_int8_precision = (
            False  # Can be toggled for hardware with INT8 tensor core support
        )
        # Optimized for the given matrix sizes (128, 512, 1024, 2048)
        # Higher threshold as the custom kernel performs better for larger matrices
        self.size_threshold = 32768  # Empirical threshold

    def forward(self, A: torch.Tensor, B: torch.Tensor) -> torch.Tensor:
        """
        Performs batched matrix multiplication with optimized CUDA implementation.

        Args:
            A: Input tensor of shape (batch_size, m, k).
            B: Input tensor of shape (batch_size, k, n).

        Returns:
            C: Output tensor of shape (batch_size, m, n).
        """
        # Fall back to PyTorch's native implementation for CPU tensors
        if not A.is_cuda or not B.is_cuda:
```

```python
        return torch.bmm(A, B)

    # For small matrices, PyTorch's native BMM might be more efficient
    matrix_size = A.size(1) * A.size(2) * B.size(2)
    if matrix_size < self.size_threshold:
        return torch.bmm(A, B)

    try:
        # Use INT8 precision for maximum throughput on compatible hardware
        if (
            self.use_int8_precision
            and torch.cuda.get_device_capability(A.device)[0] >= 7
        ):
            return self.custom_ops.batched_matmul_int8_cuda(A, B)

        # Use half precision for tensor cores if enabled and supported
        if (
            self.use_half_precision
            and torch.cuda.get_device_capability(A.device)[0] >= 7
        ):
            return self.custom_ops.batched_matmul_half_cuda(A, B)

        # Use our custom CUDA kernel for standard precision
        return self.custom_ops.batched_matmul_cuda(A, B)
    except Exception as e:
        # Graceful fallback to PyTorch implementation if custom kernel fails
        print(f"Custom kernel error: {e}, falling back to PyTorch bmm")
        return torch.bmm(A, B)
```

