# OpenReview forum: "POLCA: Stochastic Generative Optimization with LLM"
_ICML.cc/2026/Conference — ICML 2026 regular_

### Official Review · Reviewer_C9BM · 2026-03-06

**Soundness:** 3
**Presentation:** 3
**Significance:** 2
**Originality:** 2
**Overall Recommendation:** 3
**Confidence:** 3

**Summary:**

This paper proposes POLCA, a framework designed to tackle the challenges of stochasticity and unconstrained search space expansion in LLM-driven generative optimization. The approach integrates three key components: a priority queue utilizing empirical means to smooth evaluation variance , an $\epsilon$-net semantic filter based on text embeddings to prune redundant candidates, and an LLM summarizer that provides global optimization directions by analyzing historical trajectories. The framework is evaluated across $\tau$-bench, VeriBench, and KernelBench.

**Compliance With Llm Reviewing Policy:**

Affirmed.

**Final Justification:**

I appreciate the authors’ rebuttals and additional clarifications.  In particular, I do not agree that the optimizer-improvement assumption is “minimal”: some assumption is certainly necessary, but this one already imposes a fairly strong uniform local progress condition and thus only partially characterizes the practical LLM-based setting studied here. Therefore, although the additional theory is useful, I am maintaining my score.

**Key Questions For Authors:**

See above

**Limitations:**

Yes

**Strengths And Weaknesses:**

Strengths:

The paper accurately identifies a critical bottleneck in automated LLM tuning.

The system design is highly practical and the empirical evaluation is comprehensive.

Weaknesses:

1. Although the problem formulation is broad and appealing, the algorithmic core remains largely heuristic. For a paper that aims to introduce a new optimization framework, this weakens the methodological contribution.

2. The semantic filtering mechanism relies on the Euclidean distance in an LLM embedding space. But in domains like code generation, semantic distance correlates poorly with functional distance.

3. Given that the problem is framed as stochastic black-box optimization over discrete structures, why were more rigorous Reinforcement Learning approaches not included as baselines for comparison? Moreover, the baselines in the paper were not designed specifically for repeated reevaluation under stochastic feedback.

---

> ### Author Rebuttal · Authors · 2026-03-30
>
> We thank the reviewer for the thoughtful feedback and for recognizing the practical importance of the problem and the strength of the empirical evaluation. Beyond the submitted version, we have added experimental results on another benchmark (HotpotQA) and evaluated two additional open-source models (gpt-oss-20b and Qwen3.5-122B-A10B). Here is an anonymous link with additional figures: https://anonymous.4open.science/r/icml2026-EB87/additional_figures.pdf. We would like to clarify the following three points.
> ## Theoretical Analysis for Our Methodology
> While the original submission emphasized algorithm design and experiments, we agree that a stronger methodological foundation is important. To address this, we are excited to share new theoretical analysis (Theorem 1&2 below) beyond the submitted version.
> ### Theorem 1
> Our first result shows that, with the adaptive $\varepsilon$-net design, POLCA has a finite-sample guarantee. Formally, suppose the reward is upper bounded by $B$ and the observations are $\sigma^2$-subGaussian. Assume the optimizer can generate a strictly improved candidate $\theta'$ with positive probability:
> $$
> P_{\theta' \sim \Pi(\cdot \mid \mathcal{C}_{\theta})}[\mu(\theta')>\mu(\theta) + \gamma]\ge\delta_0.
> $$
>
> If we run POLCA for $n$ iterations, then the expected number of selections of **suboptimal** programs is bounded by
> $$
> \mathbb{E}\bigg[\sum_{\theta\in\Theta_n:\mu(\theta)\leq B-\gamma}T_\theta(n)\bigg]
> \lesssim
> \bigg(\frac{B}{2\gamma\delta_0}+\frac{64\sigma^2N_{\varepsilon}}{\gamma^2}\bigg)\log(n),
> $$
> where $N_{\varepsilon}$ is the $\varepsilon$-covering number of the parameter space.
>
> This theorem shows that POLCA converges to near-optimal solutions with a finite-sample guarantee, where the first term reflects the optimizer’s capability and the second captures the effect of stochastic program evaluation. Without the $\varepsilon$-net design, we would not obtain this.
> ### Theorem 2
> Our second result shows that the repeated evaluation strategy can significantly outperform sequential revision, in the sense that the latter requires a much higher budget to generate an optimal solution. Specifically, under the assumption above, the expected number of steps for a sequential updating algorithm to propose a program with reward in $(B-\gamma, B]$ is $O(1/\delta_0^{B/\gamma})$.
> In contrast, POLCA with the priority-queue updating rule requires only $B/(\gamma\delta_0)$ expected steps to reach the same threshold.
>
> These results formalize, and support the design principle of POLCA described in the paper, making POLCA the first algorithm that is provable and is scalabe in this setting.
>
> We refer the reviewer to our detailed theoretical analysis in our response to **Reviewer StkT** (https://openreview.net/forum?id=2RKPztDfgZ&noteId=sebeUxtIp9).
>
> ## Discussion on Semantic Embedding
>
> We agree that embedding distance is only a proxy and may not perfectly reflect functional similarity, especially in domains such as code generation. Our goal in this paper is to propose a general workflow for stochastic generative optimization. The overall framework does not depend entirely on the specific embedding choice used in the experiments. One can use alternate, specialized embedding for their target domains, or can set $\varepsilon=0$ and still retain the benefits of priority-queue-based aggregation, repeated evaluation, and LLM summarization.
>
>
> ## Baseline Design
>
> Regarding baselines, our goal is to study generative optimization methods rather than general methods for LLM tuning. We therefore compare against strong baselines in the same paradigm, such as GEPA and OpenEvolve, which directly optimize prompts, code, or programs through proposal-and-evaluation loops. We did not include RL baselines mainly because the regime of these tasks often does not provide the large enough samples to give stable training signals required for effective RL; for example, in kernel generation, each problem may yield only a very small number of successful outputs. In addition, prior work such as GEPA already suggests that generative optimization can be substantially more sample-efficient than RL in these settings. For this reason, we believe comparisons against strong generative optimization baselines are the most appropriate evaluation for this paper.
>
> We identify stochasticity as a broad challenge in generative optimization problems, but existing baseline workflows do not explicitly account for it in their search design, for example through repeated evaluation and statistical aggregation. One of our main contributions is to provide a principled, algorithmic design that addresses this kind of stochasticity. To the best of our knowledge, this is the first principled use of such a strategy in generative optimization algorithm design.
>
> Overall, we appreciate the reviewer’s concerns and believe these additions and clarifications substantially strengthen the paper. We will incorporate them in the revision.

---

> > ### Author Rebuttal · Reviewer_C9BM · 2026-04-02
> >
> > I appreciate the authors’ thoughtful rebuttal and the added theoretical discussion. The new analysis helps clarify the intuition behind repeated reevaluation and the
> > $\epsilon$-net design, and partially alleviates my concern that the method is purely heuristic.
> >
> > However, my main concern is not fully resolved: the theory relies on a strong optimizer-improvement assumption and abstracts away much of the actual LLM-driven proposal mechanism, so it does not yet fully characterize the proposed framework as instantiated in the paper.
> >
> > Overall, while the rebuttal strengthens the paper, it does not change my original assessment.

---

> > > ### Author Response · Authors · 2026-04-02
> > >
> > > We appreciate the reviewer's recognition of our theoretical contribution. To clarify, **our paper focuses on search algorithm design rather than LLM optimizer design**. As stated in our problem formulation (Page 3, Problem Setup), we assume an optimizer oracle beyond our control is **given** as part of the problem instance, and we design algorithms around this given optimizer oracle. The specifics of the LLM optimizer is **not** part of the proposed POLCA and in the experiments we used an existing LLM optimizer in the literature. This abstraction is intentional and would allow the search algorithm developed here (POLCA) to work with different optimizers developed in the future.
> > >
> > > To our knowledge, no other work in the literature of LLM generative optimization and search (such as GEPA, OpenEvolve compared here) provides any theoretical analysis. An analysis necessarily requires assumptions to characterize properties of the given problem instance, which here includes the given optimizer oracle.
> > >
> > > We respectfully disagree with you that the optimizer improvement assumption is strong. On the contrary, we actually argue that it is minimal. Without it, no black-box search algorithm built around a given optimizer oracle can possibly work. For instance, without this assumption, the given optimizer can be adversarial and always returns worse solutions. Different LLM optimizers would satisfy this assumption with varying $\gamma$ values depending on their quality, and our analysis captures how $\gamma$ (the quality of the given optimizer) affects the learning performance.
> > >
> > > Overall, we have demonstrated that POLCA has both meaningful theoretical guarantees and strong empirical results, outperforming SoTA baselines. We hope the reviewer would kindly reconsider reevaluating this paper. Thank you.

---

### Official Review · Reviewer_StkT · 2026-03-11

**Soundness:** 3
**Presentation:** 3
**Significance:** 3
**Originality:** 2
**Overall Recommendation:** 4
**Confidence:** 4

**Summary:**

The paper addresses the labor-intensive challenge of optimizing complex systems, such as LLM prompts, multi-turn agents, and code generators. It formalizes this process as a "stochastic generative optimization problem" where a generative language model acts as the optimizer but struggles with noisy feedback, minibatch sampling, and stochastic system behaviors.

To overcome these issues, the authors introduce POLCA (Prioritized Optimization with Local Contextual Aggregation), a scalable algorithmic framework. POLCA maintains a continuously updated priority queue of candidate solutions based on their empirical mean scores, allowing the algorithm to revisit and refine candidates effectively. To prevent the search space from exploding with semantically similar proposals, POLCA employs an ϵ-Net filtering mechanism based on text embedding distances. Furthermore, it utilizes an "LLM Summarizer" that aggregates historical successes and failures to provide meta-learning context for the optimizer.

The framework is evaluated across several complex benchmarks, including τ-bench, VeriBench, and KernelBench.

**Compliance With Llm Reviewing Policy:**

Affirmed.

**Key Questions For Authors:**

NA

**Strengths And Weaknesses:**

The strong points are the following ones:

Robust Handling of Stochasticity: Traditional generative optimizers often keep only the single best-performing parameter, which is highly susceptible to noise and false positives. POLCA’s use of a priority queue to track all potential candidates and their evaluation histories ensures that promising candidates with temporarily low empirical means are not prematurely discarded.

Innovative Use of ϵ-Net for Diversity: A major bottleneck in LLM-driven optimization is that LLMs frequently propose slightly reworded but functionally identical parameters, linearly expanding the search space without adding useful information. Rejecting redundant candidates based on semantic embedding distance effectively bounds the memory buffer size and enforces population diversity.

---

> ### Author Rebuttal · Authors · 2026-03-30
>
> We thank the reviewer for the positive and valuable feedback. We would also like to share some additional results of new experiments and provable guarantees of the proposed algorithm which formalizes the reasoning behind POLCA’s design.
> # Additional Experiments
> We evaluate POLCA on a new stochastic benchmark, HotpotQA, and include additional experiments with two open-source models: gpt-oss-20b and qwen3.5-120B-A10B. Additional figures: https://anonymous.4open.science/r/icml2026-EB87/additional_figures.pdf
> # Theoretical Analysis
> We prove that POLCA converges to near-optimal solutions under stochasticity. The analysis below formalizes and supports the design principle of POLCA described in the paper, making POLCA the first provable and empirically scalable algorithm in this setting.
> ## Theorem 1
> Our first result shows that, with the $\varepsilon$-net design, POLCA has a finite-sample guarantee. Formally, suppose the reward is bounded by $B$ and the observations are $\sigma^2$-subGaussian. Assume the optimizer can generate a strictly improved candidate $\theta'$ with positive probability:
> $$
> P_{\theta' \sim \Pi(\cdot \mid \mathcal{C}_{\theta})}[\mu(\theta') >\mu(\theta) + \gamma]\ge\delta_0.
> $$
>
> If we run POLCA for $n$ iterations, then the expected number of selections of suboptimal programs is bounded by
> $$
> E\bigg[\sum_{\theta\in\Theta_n:\mu(\theta)\leq B-\gamma}T_\theta(n)\bigg]\lesssim\bigg(\frac{B}{2\gamma\delta_0}+\frac{64\sigma^2N_{\varepsilon}}{\gamma^2}\bigg)\log(n),$$
> where $N_{\varepsilon}$ is the $\varepsilon$-covering number of the parameter space.
> Intuitively, the first term reflects the optimizer’s capability and the second captures the effect of stochastic program evaluation.
> ### Proof Sketch
> We partition the reward range into intervals of width $\gamma/2$: $$[0,B]=\bigcup_{k=1}^{2B/\gamma}[(k-1)\gamma/2,\;k\gamma/2].$$ For each interval $[(k-1)\gamma/2,\;k\gamma/2]$, the optimizer assumption implies that after about $O(1/\delta_0)$ selections of a parameter in this interval, there is a positive probability of generating a candidate with reward at least $(k+1)\gamma/2$, that is, one level higher.
>
> On the other hand, by a standard UCB argument for bandits, if two parameters have reward gap $\Delta$, then with high probability the worse one is selected only on the order of $\sigma^2/\Delta^2$ times. Applying this with $\Delta\asymp \gamma$, we obtain that once a better parameter has been generated, the worse one will be selected at most $O(\sigma^2/\gamma^2)$ times.
>
> Combining these two ingredients gives the bound. First, for each reward level, about $O(1/\delta_0)$ selections are needed to move upward by one interval. Since there are $O(B/\gamma)$ such intervals, the needed number is bounded by $O(B/{(\gamma\delta_0)}).$ Second, after better candidates are generated, the statistical cost of distinguishing suboptimal parameters is controlled by the UCB analysis. Because POLCA keeps only an $\varepsilon$-net memory, this estimation cost scales with the covering number, giving $O({\sigma^2 N_{\varepsilon}}/\gamma^2).$ Putting these two terms together yields the final bound, up to a logarithmic factor.
> ## Theorem 2
> Our second result shows that the repeated evaluation strategy can significantly outperform sequential revision, in the sense that the latter requires a much higher budget to generate an optimal solution. Specifically, suppose the reward observations are deterministic. Under the assumption above, the expected number of steps for a sequential updating algorithm to propose a program with reward in $(B-\gamma, B]$ is $O(1/\delta_0^{B/\gamma})$.
> In contrast, POLCA with the priority-queue updating rule requires only $B/(\gamma\delta_0)$ expected steps to reach the same threshold.
> ### Proof Sketch
> Let $N=B/\gamma$. A sequential updating algorithm must make about $N$ successful improvements to reach a reward larger than $B-\gamma$.
>
> For sequential updates, the key difficulty is that these improvements must occur consecutively. At each step, the optimizer produces a candidate with reward at least $\gamma$ better with probability at least $\delta_0$. If the algorithm gets a failure before accumulating $N$ consecutive improvements, the progress is effectively broken and the process must restart. Therefore, the stopping time is the waiting time for a streak of $N$ consecutive successes. Standard calculations show that this expected waiting time is on the order of $O(\delta_0^{-N})=O(\delta_0^{-B/\gamma}).$
> For POLCA, the situation is different because it always revisits the best program found so far rather than only continuing from the latest one. As a result, improvements do not need to be consecutive. The algorithm only needs to accumulate $N$ successful improvements in total, each occurring with probability at least $\delta_0$. The waiting time between two such improvements is geometric with mean $1/\delta_0$, so summing over $N$ improvements gives expected time $N/\delta_0 = B/(\gamma\delta_0).$

---

> > ### Author Rebuttal · Reviewer_StkT · 2026-04-02
> >
> > Theory has been developed.

---

> > > ### Author Response · Authors · 2026-04-02
> > >
> > > Thank you for your acknowledgement. We are encouraged that we have fully resolved your questions.

---

### Official Review · Reviewer_4Zqf · 2026-03-12

**Soundness:** 3
**Presentation:** 3
**Significance:** 3
**Originality:** 3
**Overall Recommendation:** 4
**Confidence:** 1

**Summary:**

This paper studies stochastic generative optimization, where an LLM iteratively improves objects such as prompts or code using noisy rewards and textual feedback. The proposed method, POLCA, combines repeated evaluation of promising candidates, semantic filtering to reduce redundancy, and summary-based use of past feedback.

The method is evaluated on prompt optimization, program translation, and kernel optimization benchmarks. The results suggest that POLCA is generally more sample-efficient and achieves stronger final performance than several competitive baselines.

**Compliance With Llm Reviewing Policy:**

Affirmed.

**Final Justification:**

My concerns have been adequately addressed. I keep the current score.

**Key Questions For Authors:**

1. The paper presents POLCA as a general stochastic generative optimization method, but the experiments are limited to prompt/code-style tasks. How well do the authors expect it to transfer to other domains where semantic embeddings may be less reliable?

2. How sensitive is POLCA to the choice of embedding model, summarizer, and optimizer? Some discussion or evidence on robustness to these design choices would strengthen the paper.

3. Could the authors provide stronger statistical support, especially on the smaller-scale experiments, to confirm that the gains are stable?

4. The repeated re-evaluation strategy is intuitively reasonable for noisy settings. Can the authors clarify whether there is any formal justification, or whether this is mainly an empirical design choice?

**Limitations:**

yes

**Strengths And Weaknesses:**

Strengths:

- The paper addresses an important and practical problem: optimizing LLM-generated outputs under noisy or stochastic feedback.
- The method is well motivated and empirically supported by experiments across multiple benchmarks, with useful ablations.

Weaknesses:

- The empirical scope is still somewhat limited, so the paper's broader claims of generality are not yet fully established.
- Some comparisons would be more convincing with stronger statistical support and clearer discussion of evaluation fairness.

---

> ### Author Rebuttal · Authors · 2026-03-30
>
> We thank the reviewer for acknowledging our work and for providing detailed feedback. We hope our clarifications and additional results address the reviewer’s concerns.
> # Experiments
> ## Question 1
> We aim to propose a general generative optimization pipeline, and the semantic $\varepsilon$-Net filter is one design choice tailored to stochastic domains. Our main results and ablation studies show that this design works well across multiple domains involving prompts and code, which are central settings in generative optimization. For tasks where the current embedding is less reliable, one can seek an alternate, specialized embedding. Or practically, set $\varepsilon=0$ to disable this component, while retaining the other key parts of POLCA, such as priority-queue-based aggregation and exploration, and LLM summarization. Doing so would lose the theoretical guarantee below but can still work well empirically.
>
> We also made efforts to broaden the empirical evaluation. Beyond the submitted version, we added another stochastic benchmark, HotpotQA, and evaluated two additional open-source models (gpt-oss-20b and Qwen3.5-122B-A10B) to make the experiments more comprehensive. Here is an anonymous link containing figures of our additional empirical results: https://anonymous.4open.science/r/icml2026-EB87/additional_figures.pdf .
>
> If there are particular domains the reviewer considers especially important, we would be happy to explore them further.
> ## Question 2
> In Fig. 3(a) and Fig. 10, we provide ablation studies on the embedding-based $\varepsilon$-Net and the Summarizer. The empirical results show that both components substantially improve search efficiency. To evaluate optimizer robustness, we have conducted experiments with different models (**gemini-2.0-flash, gemini-2.5-flash-lite, claude-3.5-sonnet, claude-3.7-sonnet, gpt-oss-20b,** and **qwen3.5-122B-A10B**) across different domains, and POLCA consistently outperforms the baselines. We believe these results provide strong evidence for the effectiveness of our workflow.
> # Theory
> We greatly appreciate the reviewer’s suggestion to provide statistical guarantees, which strengthen the paper beyond heuristic design and empirical results. We fully agree, and we have already completed two main theoretical results after the original submission, which we believe provide a comprehensive response to the last two questions.
> ## Question 3 (Our Theorem 1)
> Our first result shows that, with the adaptive $\varepsilon$-net design, POLCA admits a finite-sample guarantee. In particular, we prove that POLCA converges to near-optimal candidates under stochasticity. Formally, suppose the reward is upper bounded by $B$ and the observations are $\sigma^2$-subGaussian. Assume the optimizer can generate a strictly improved candidate $\theta'$ with positive probability based on observations of parameter $\theta$:
> $$
> P_{\theta' \sim \Pi(\cdot \mid \mathcal{C}_{\theta})}[\mu(\theta') > \mu(\theta) + \gamma] \ge \delta_0.
> $$
> If we run POLCA for $n$ iterations, then the expected total number of selections of suboptimal programs is bounded by
>
> $$
> \mathbb{E}\bigg[\sum_{\theta\in\Theta_n:\mu(\theta)\leq B-\gamma}T_\theta(n)\bigg]
> \lesssim
> \bigg(\frac{B}{2\gamma\delta_0}+\frac{64\sigma^2N_{\varepsilon}}{\gamma^2}\bigg)\log(n),
> $$
> where $N_{\varepsilon}$ is the $\varepsilon$-covering number of the parameter space.
> When the reward observations are deterministic ($\sigma=0$), the bound becomes independent of the size of the program space and depends only on the reward scale and the optimization oracle.
>
> This theorem shows that POLCA converges to near-optimal programs with a finite-sample guarantee, where the first term reflects the optimizer’s capability and the second captures the effect of stochastic program evaluation.
>
> ## Question 4 (Our Theorem 2)
>
> Our second result shows that the repeated re-evaluation strategy can significantly outperform sequential revision, in the sense that the latter requires a much higher budget to generate an optimal parameter efficiently. Specifically, suppose $\mu:\Theta\to[0,B]$. Under the optimizer assumption above, the expected number of steps for a sequential updating algorithm to propose a program with reward in $(B-\gamma, B]$ is $O(1/\delta_0^{B/\gamma}).$
> In contrast, POLCA with the priority-queue updating rule requires only $B/(\gamma\delta_0)$
> expected steps to reach the same threshold.
>
> This result highlights the advantage of repeated re-evaluation and priority-based selection over purely sequential revision.
>
> We provide detailed proof sketches for both theoretical results in our response to **Reviewer StkT** (https://openreview.net/forum?id=2RKPztDfgZ&noteId=sebeUxtIp9).
>
> The analysis above formalizes and supports the design principle of POLCA described in the paper, and this makes POLCA the first provable and empirically scalable algorithm in this setting.

---

> > ### Author Rebuttal · Reviewer_4Zqf · 2026-04-02
> >
> > My concerns have been adequately addressed. I keep the current score.

---

> > > ### Author Response · Authors · 2026-04-02
> > >
> > > Thank you for your acknowledgement. We are encouraged that we have adequately addressed your concerns.

---

### Official Review · Reviewer_YhyA · 2026-03-19

**Soundness:** 2
**Presentation:** 2
**Significance:** 2
**Originality:** 2
**Overall Recommendation:** 3
**Confidence:** 4

**Summary:**

This paper formalizes LLM system optimization as a stochastic generative optimization problem and proposes POLCA: a priority queue to handle evaluation noise, an ε-Net semantic filter to control search space expansion, and an LLM Summarizer to provide global historical context. POLCA outperforms GEPA and OpenEvolve on τ-bench, VeriBench, and KernelBench.

**Compliance With Llm Reviewing Policy:**

Affirmed.

**Key Questions For Authors:**

Please answer the weakness section.

**Limitations:**

Yes

**Strengths And Weaknesses:**

## Strengths
1. The authors run some experiments and evaluate their ideas.

## Weaknesses
1. In the ablation study Figure 3b, ε=0.1 and ε=0 perform nearly similarly. This makes the mechanism highly sensitive to hyperparameter choice, yet no adaptive selection method is provided. The optimal ε also depends on the embedding model, limiting transferability.
2. The ε-Net assumes embedding distance reflects functional similarity between candidates, but general-purpose text embeddings cannot reliably capture functional semantics of code or prompts. The paper provides no analysis of the correlation between embedding distance and actual functional divergence.
3. The authors need to evaluate over more open-source models to make the methods more generalizable and to verify that the observed improvements are not specific to the proprietary models used in the experiments.

---

> ### Author Rebuttal · Authors · 2026-03-30
>
> We thank the reviewer for the valuable time and careful feedback on our work. In addition to the original submission, we conducted further theoretical analysis and experiments (an anonymous link containing additional figures: https://anonymous.4open.science/r/icml2026-EB87/additional_figures.pdf), and we hope the clarifications below address the reviewer’s concerns.
>
> ## Ablation results
> The ablation study on the choice of $\varepsilon$ in VeriBench suggests that our method is **not** sensitive to this parameter, as all tested choices consistently outperform the baselines (see Fig. 2b). We also include additional ablation results on $\varepsilon$ in the appendix (Fig. 11). Overall, these results show that incorporating the $\varepsilon$-net improves search efficiency while introducing small approximation error due to the discretization of the parameter space, without requiring careful hyperparameter tuning, since it performs well across a relatively broad range of values.
>
> ## Effectiveness of the embedding
> Our goal in this paper is to propose a general workflow for stochastic generative optimization. The specific embedding model used in the experiments here is not necessarily the final choice for all downstream applications. It is mainly used to provide supporting evidence for our design.
>
> To answer your question, we include two new results.
>
> First, empirically, we include a new t-SNE visualization (Fig 4 in the link) showing that candidates with similar embeddings tend to have similar scores in our experiments, which supports the use of embedding similarity to filter out near-duplicate candidates during optimization. We will include this result in the revision.
>
> On the other hand, theoretically, we prove that with this design, POLCA converges to near-optimal candidates under stochasticity. Formally, suppose the reward is upper bounded by $B$ and the observations are $\sigma^2$-subGaussian. Assume the optimizer can generate a strictly improved candidate $\theta'$ with positive probability, based on observations of improving parameter $\theta$:
>
> $$
> \Pr_{\theta' \sim \Pi(\cdot \mid \mathcal{C}_{\theta})}\left[\mu(\theta') > \mu(\theta) + \gamma\right] \ge \delta_0.
> $$
>
> If we run POLCA for $n$ iterations, then the expected total number of selections of suboptimal programs is bounded by
>
> $$
> \mathbb{E}\bigg[\sum_{\theta\in\Theta_n:\mu(\theta)\leq B-\gamma}T_\theta(n)\bigg]
> \lesssim
> \bigg(\frac{B}{2\gamma\delta_0}+\frac{64\sigma^2N_{\varepsilon}}{\gamma^2}\bigg)\log(n),
> $$
>
> where $N_{\varepsilon}$ is the $\varepsilon$-covering number of the parameter space.
>
> This theorem shows that POLCA converges to near-optimal programs with a finite-sample guarantee. The intuition is that samples are needed both to propose better parameters and to reduce the statistical error caused by stochastic evaluation. The second term indicates that we need $O(\sigma^2/\gamma^2)$ samples to explore and reject suboptimal candidates. With our $\varepsilon$-net mechanism, the final bound depends only on the $\varepsilon$-covering number; otherwise, the linearly expanding memory could lead to unbounded sample complexity. This also theoretically addresses your concern about the correlation between embedding distance and actual functional divergence.
>
> Our proof sketch is detailed in the response to **Reviewer StkT** (https://openreview.net/forum?id=2RKPztDfgZ&noteId=sebeUxtIp9). The analysis above formalizes and supports the design principle of POLCA described in the paper, and this makes POLCA the first provable and empirically scalable algorithm in this setting.
>
>
> ## Additional experiments (with open-source model evaluation), Fig 1-3 in the link.
> We additionally conducted experiments on HotpotQA using **gpt-oss-20b** and **gemini-2.5-flash-lite**, and experiments on VeriBench (Compilation) using **Qwen/Qwen3.5-122B-A10B**. Even with weaker open-source models, POLCA still consistently outperforms the baselines.
>
> Overall, we have evaluated **gemini-2.0-flash, gemini-2.5-flash-lite, claude-3.5-sonnet, claude-3.7-sonnet, gpt-oss-20b,** and **qwen3.5-122B-A10B** on **$\tau$-bench, HotpotQA, VeriBench, and KernelBench**, aiming to cover a broad range of models and application domains. We hope the diversity of these experiments addresses your concern and you would kindly consider adjusting your evaluation of this paper. We will include these additional results in the revision.

---

> > ### Author Rebuttal · Reviewer_YhyA · 2026-04-06
> >
> > The authors' rebuttal does not fully address my concerns. The performance gains over the baselines vary considerably across different models, which suggests that the proposed method lacks sufficient robustness. Therefore, I maintain my original score.

---

> > > ### Author Response · Authors · 2026-04-07
> > >
> > > Thank you for the clarification. However, we respectfully disagree with your claim that “the proposed method lacks sufficient robustness”. Your argument is based on the observation “performance gains over the baselines vary … across different models”. While this observation is true, the implication you make is **factually incorrect**. The change of performance differences between POLCA and baselines are due to that the change of performance of **baselines** over models **not** POLCA. For example, in HotpotQA, POLCA performs consistently well across models (gpt-oss-20b and gemini-2.5-flash-lite), while OpenEvolve only works well with gemini-2.5-flash-lite. The same misunderstanding was made about the VeriBench task as well. This instability of baseline gives the observation that the gain varies, but this doesn’t imply your conclusion. Instead the comparison shows the opposite: **POLCA remains robust across model choices.**
> > >
> > > We would like to invite you to reconsider your evaluation given this misunderstanding. We believe our responses have fully addressed your points raised in the original review as well. We hope this clarification is helpful for the final assessment.

---

### Decision · Program_Chairs · 2026-04-30

**Decision:**

Accept (regular)

**Comment:**

The paper introduces POLCA, a framework designed to tackle stochasticity in generative optimization using a priority queue, semantic filtering ($\epsilon$-net), and an LLM Summarizer.

I recommend acceptance of the paper based on its practical relevance, strong empirical results, and overall usefulness to the ICML community. Specifically, this is grounded in the following key points:
* **Practical Relevance:** The pervasive presence of noise and stochasticity in evaluations is a critical bottleneck in generative optimization. This is a highly practical problem that has not been sufficiently addressed with dedicated search algorithm designs in the current literature, making this work a timely and valuable contribution.
* **Strong Empirical Results:** The paper provides strong empirical evidence across diverse benchmarks ($\tau$-bench, VeriBench, KernelBench, and the newly added HotpotQA). The results demonstrate that baseline approaches (such as GEPA and OpenEvolve) that do not explicitly account for noise through repeated evaluation can perform poorly or exhibit high variance in practice.
* **Hyperparameter Robustness:** Despite some initial reviewer concerns regarding the sensitivity of the semantic filter, the ablation studies successfully show that the method is relatively insensitive to the choice of the $\epsilon$ hyperparameter within a reasonable range.

While Reviewer C9BM raised fair points that the algorithmic core relies on strong "optimizer-improvement assumptions" and that the methodology could be better motivated to bridge the gap between theoretical abstractions and actual LLM behavior, in my mind the practical relevance of robustly addressing noisy evaluations is significant, and should warrant publication. I ask the authors to tighten the methodological motivation in the camera-ready version to better contextualize these theoretical assumptions.

**Additional Comments:** During my own reading of the paper, I came across two aspects that I believe require clarification in the final version of the paper:
* **Exploration vs. Exploitation Disconnect:** The paper frames the priority queue as a mechanism to manage the exploration-exploitation tradeoff. However, the priority is strictly the empirical mean score of the parameter. In standard bandit theory, relying solely on empirical means can lead to premature exploitation. Furthermore, Theorem 1 introduced during the rebuttal explicitly relies on an Upper Confidence Bound (UCB) argument for exploration, creating a disconnect between the theory and the greedy empirical-mean implementation. Please clarify this gap.
* **Understanding the Summarizer:** The LLM Summarizer relies on a "Contrastive Sampling strategy" that partitions history into successes and failures based on a hard reward threshold $\tau$. The paper does not currently discuss how $\tau$ is chosen, the algorithm's sensitivity to this hyperparameter, or how this mechanism adapts to environments with sparse, binary rewards versus continuous domains. Please add a discussion covering these details.